# Benchmarking Multimodal Retrieval Augmented Generation with Dynamic VQA Dataset and Self-adaptive Planning Agent

**Yangning Li**[1,2]*, **Yinghui Li**[1]*, **Xinyu Wang**[3‡], **Yong Jiang**[3‡], **Zhen Zhang**[3], **Xinran Zheng**[4]
**Hui Wang**[2], **Hai-Tao Zheng**[1,2‡], **Fei Huang**[3], **Jingren Zhou**[3], **Philip S. Yu**[5]
[1]Shenzhen International Graduate School, Tsinghua University [2]Peng Cheng Laboratory
[3]Tongyi Lab, Alibaba Group [4]University College London [5]University of Illinois Chicago

## Abstract

Multimodal Retrieval Augmented Generation (mRAG) plays an important role in mitigating the "hallucination" issue inherent in multimodal large language models (MLLMs). Although promising, existing heuristic mRAGs typically predefined fixed retrieval processes, which causes two issues: (1) Non-adaptive Retrieval Queries. (2) Overloaded Retrieval Queries. However, these flaws cannot be adequately reflected by current knowledge-seeking visual question answering (VQA) datasets, since the most required knowledge can be readily obtained with a standard two-step retrieval. To bridge the dataset gap, we first construct Dyn-VQA dataset, consisting of three types of "dynamic" questions, which require complex knowledge retrieval strategies variable in query, tool, and time: (1) Questions with rapidly changing answers. (2) Questions requiring multi-modal knowledge. (3) Multi-hop questions. Experiments on Dyn-VQA reveal that existing heuristic mRAGs struggle to provide sufficient and precisely relevant knowledge for dynamic questions due to their rigid retrieval processes. Hence, we further propose the first self-adaptive planning agent for multimodal retrieval, **OmniSearch**. The underlying idea is to emulate the human behavior in question solution which dynamically decomposes complex multimodal questions into sub-question chains with retrieval action. Extensive experiments prove the effectiveness of our OmniSearch, also provide direction for advancing mRAG.

## 1 Introduction

Multimodal Retrieval Augmented Generation (mRAG) (Zhao et al., 2024; 2023; Gao et al., 2023) aims to provide more comprehensive, accurate and up-to-date knowledge from external sources for AI systems. It has emerged as a key technology to mitigate the "hallucination" issue (Liu et al., 2024a; Bai et al., 2024) inherent in multimodal large language models (MLLMs).

Existing heuristic mRAG methods typically predefined fixed retrieval processes that ground all modalities into one primary modality (usually text), then retrieve for a single time. Despite the promising results, these retrieval strategies suffer from two issues: **(1) Non-adaptive Retrieval Queries** refer to the fixed

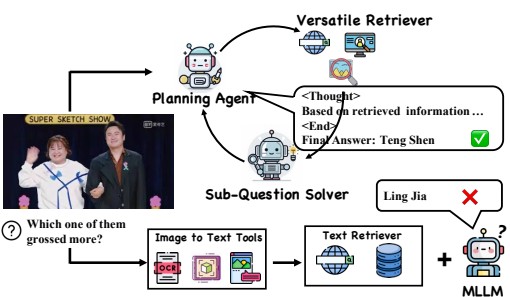

Figure 1: Bottom: Heuristic mRAG based VQA. Upper: OmniSearch based VQA.

retrieval processes and query structures of heuristic mRAG methods. These inflexible retrieval strategies fail to adapt to evolving contexts or intermediate findings within a question, hindering the model from re-retrieving to further comprehend, verify, or rethink the question. For example, in

---

*Equal Contribution. ‡Corresponding Author. This work was led by Xinyu Wang at Alibaba Group. The code and dataset is open-sourced at `https://github.com/Alibaba-NLP/OmniSearch`.

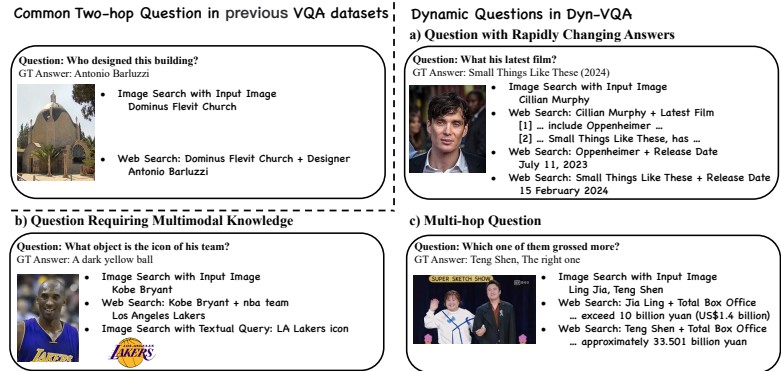

Figure 2: Dynamic VQA examples that require different kinds of retrieval strategies.

Figure 2, question (a) asks, "What is his (Cillian Murphy's) latest film?" A fixed retrieval process returns multiple relevant films, but heuristic methods fail to construct further retrieval based on the retrieved content to distinguish between the sequence of different films. **(2) Overloaded Retrieval Queries** refer to heuristic retrieval methods refer to heuristic mRAG methods merely format a single query by concatenating textual descriptions of objects in images with input questions. A single query carries multiple retrieval aspects, leading to ambiguous retrieval and influx of superficially relevant knowledge yet not essential to the question solving. For example, in Figure 2, question (c) asks, "Which one of them (two actors, Ling Jia, and Teng Shen) grossed more?" Heuristic methods might generate a single query like "Ling Jia, Teng Shen, Which one of them grossed more?", which contains the intent to retrieve box office information for both actors. This mixed query conversely fails to provide precise knowledge for each individual aspect. Therefore, as shown in Figure 1, when faced with real-world questions requiring complex knowledge, current heuristic mRAG methods fail to provide sufficient and precise knowledge, due to their ***rigidity issues***.

Unfortunately, although several knowledge-seeking visual question answering (VQA) datasets (Chen et al., 2023; Schwenk et al., 2022) are widely utilized as mRAG benchmarks, they cannot adequately reflect the rigidity issues of heuristic mRAGs in acquiring complex knowledge. Since most questions in them merely require textual knowledge within two-hop, which can be readily obtained by heuristic mRAGs with a standard two-step retrieval process. As illustrated in the upper left of Figure 2, the most common type of question inquires about a certain property of an object.

To bridge the mRAG dataset gap, we first construct a challenging dataset, Dyn-VQA, comprising 1,452 dynamic questions that require complex multimodal knowledge retrieval for solution. ***Dynamic questions*** are defined as questions that require the model to flexibly provide knowledge retrieval solutions, where the query, tool, and time of retrievals are all variable. These questions cannot be solved by a predefined retrieval process. Concretely, there types of dynamic questions are included in Dyn-VQA: **(1) Questions with rapidly changing answers.** Since the context knowledge of such question updates frequently, the retrieved content may be mixed with outdated and newer knowledge that is difficult to distinguish. This requires mARG methods to flexibly plan additional retrievals based on feedback from current retrieved content for further comprehension, rather than merely a one-time retrieval. **(2) Questions requiring multi-modal knowledge.** The knowledge necessary by Dyn-VQA spans various modalities. This demands that mRAG methods retrieve knowledge across diverse modalities with tailored retrieval APIs, differing from most VQA datasets limited in seeking textual knowledge with multimodal questions. **(3) Multi-hop questions.** Questions in Dyn-VQA necessitate varied reasoning hops for solution, which entails that mRAG methods conduct various retrieval steps. While existing VQA datasets primarily focus on two-hop question, i.e., identifying visual concepts via text and then answering single-hop textual question.

We evaluated the performance of various mRAG methods combined with leading MLLMs on Dyn-VQA. Experiments reveal that existing heuristic mRAGs struggle to provide sufficient and precisely relevant knowledge for dynamic questions in Dyn-VQA, due to their rigid retrieval processes.

To address these issues, we further propose a self-adaptive planning agent for multimodal retrieval, **OmniSearch**[1]. The underlying idea is to emulate the human behavior in question solution which dynamically decomposes complex multimodal questions into sub-question chains with retrieval ac-

---

[1] We aim for OmniSearch to achieve Omnipotent Multimodal Search, solving real-world issues in future.

tion. At each step, OmniSearch flexibly adjusts the next action according to question-solving state and retrieved content, with diverse purposes such as deepening comprehension of retrieved content, modifying retrieval method for current sub-question, proposing the next sub-question, etc. It is noteworthy that OmniSearch can serve as a plug-and-play RAG module, cooperating with arbitrary MLLMs to augment their capability in addressing complex dynamic questions. Two different versions of OmniSearch are developed based on closed-source GPT-4V (Achiam et al., 2023) and open-source Qwen-VL-Chat (Bai et al., 2023a), respectively. As far as we know, OmniSearch is the first multimodal retrieval agent for VQA tasks with self-adaptive planning and scalable submodules.

In summary, our main contributions are fourfold:

- We reveal that current VQA-based mRAG benchmarks don't reflect real-world needs for dynamic knowledge retrieval and propose the Dyn-VQA dataset with three types of dynamic questions.

- We benchmark various mRAG methods with leading MLLMs on Dyn-VQA, demonstrating their flaw in providing sufficient and relevant knowledge for dynamic questions.

- We propose OmniSearch, a self-adaptive retrieval agent that plans each retrieval action in real-time according to question solution stage and current retrieval content.

- Extensive experiments prove the effectiveness of our OmniSearch. Detailed analyses are conducted to provide direction for advancing mRAG.

## 2 RELATED WORKS

### 2.1 MULTIMODAL RETRIEVAL AUGMENTED GENERATION

The mRAG methods (Zhao et al., 2023; 2024; Gao et al., 2023) aim to provide MLLMs (Lu et al., 2024b; Ye et al., 2024; Liu et al., 2024b; Du et al., 2022; Chai et al., 2024a) with more comprehensive, accurate and up-to-date world knowledge from external sources (Wu et al., 2024; Ji et al., 2024; Hou et al., 2024). They have been empirically proven to be effective on various VQA datasets, which can be categorized into two classes based on the retrieval method.

One category employs visual encoding model for direct image representation, and then retrieves the knowledge from knowledge base with the highest feature similarity. For instance, KAT (Gui et al., 2022) and Revive (Lin et al., 2022) both use the image encoder of CLIP (Radford et al., 2021) for retrieval. The other category (Hu et al., 2023; Yang et al., 2022; Lin et al., 2024) first transforms the input image into textual representation utilizing off-the-shelf tools and then performs text retrieval. For example, RA-VQA (Lin & Byrne, 2022) and RA-VQA-v2 (Lin et al., 2024) first use existing object detection and image captioning models to convert images to text, and than perform dense passage retrieval to fetch relevant text documents. Several studies (Yao et al., 2023; Xu et al., 2023) have preliminarily explored into agentic RAG, but their primary focus was on the text domain. Chen et al. (2024a) proposed a causality-enhanced agent framework focus on unimodal biases in MLLMs, while it lacks plug-and-play capabilities.

The purpose of OmniSearch aligns with previous works to furnish pertinent and accurate knowledge for MLLMs, but diverging in three aspects: (1) OmniSearch plans multiple retrieval actions for the each question with diverse retrieval tools, supplementing the missing knowledge of each modality. (2) OmniSearch dynamically adjusts subsequent retrieval actions based on retrieved content, diverging from methods formulate query solely with input questions and images. (3) OmniSearch's retrieval scope extends to the entire Internet, offering intricate yet more comprehensive knowledge.

## 3 DYN-VQA DATASET

In this section, we curated a novel dataset, Dyn-VQA, designed to evaluate the performance of existing mRAG methods in addressing questions requiring dynamic retrieval.

### 3.1 DYN-VQA CONSTRUCTION

To guarantee the dataset quality, we explain the overall goal of the dataset to the annotators, who are all professional AI researchers. A straightforward construction strategy might directly request

the annotators to write more visual questions after showing them the various types of VQA cases in Figure 2. However, in the preliminary annotation, we found that this overloaded single-step strategy is quite impractical. Annotators often found themselves in a dilemma of searching an image first, then laboriously devising a corresponding question while keeping various criteria, e.g., changing speed, and reasoning steps in their mind. Therefore, we optimize the strategy and divide them into three steps:

**Step 1. Textual Question Writing.** First, annotators are required to craft textual questions and categorize them using a three-dimensional schema based on answer update frequency (fast, slow, never), whether requiring external visual knowledge (yes, no), and reasoning steps ($\leq$ 2-hop, $>$ 2-hop). The annotation of fast or slow is determined by whether the updating occurs on yearly basis. Whether seeking visual knowledge beyond input images is also considered to separate questions emphasizing on different modalities. Meanwhile, multi-hop questions are delineated by a 2-hop cutoff, as previous VQA datasets typically focus on 2-hop. Annotators are prompted to compose questions incorporating newly emerged concepts from the past six months. The annotation difficulty is significantly reduced since visual information is not considered. Besides, English QA instances from FreshQA (Vu et al., 2023) are also included.

**Step 2. Multimodal Rewriting.** The annotator transforms textual questions from the first step into multimodal ones by replacing visual concepts with co-references (e.g., "Kobe Bryant" to "this player") and pairing the revised question with a relevant image found on Google. Images sourced from commonly used pre-trained corpus such as Wikipedia and Baidu Encyclopedia are forbidden.

**Step 3. Chinese-English Translation.** This step aligns Chinese and English parts of the Dyn-VQA for side-by-side language comparison. Chinese and English VQA instances are translated into each other using Google Translate API, followed by manual checks and corrections to guarantee accuracy, especially for proper nouns. Instances that are intractable to translate or not applicable to Chinese/English contexts are filtered. Additionally, each question is annotated with the golden query, which simplifies the question by focusing solely on the last-hop question. This means that references to visual concepts and complex intermediate reasoning are omitted from the questions.

## 3.2 DYN-VQA ANALYSIS

Table 1: Comparison of knowledge-seeking VQA datasets.

| Dataset | Knowledge Type | Ans. Change Freq. | Reasoning Step | External Visual-Seek | Human Annotation | # {I, Q, A} | Len. of Q/A | Lang. | Const. Year |
|---|---|---|---|---|---|---|---|---|---|
| VQAv2(Goyal et al., 2017) | common | never change | $\leq$ 2-hop | ✗ | ✓ | 614K | 6.2/1.1 | en | 2017 |
| OK-VQA (Marino et al., 2019) | factoid | never change | 2-hop | ✗ | ✓ | 14K | 8.1/1.3 | en | 2019 |
| S3VQA (Jain et al., 2021) | factoid | never change | 2-hop | ✗ | ✗ | 7.5K | 12.7/2.8 | en | 2021 |
| ViQuAE (Lerner et al., 2022) | fixed kb | never change | 2-hop | ✗ | ✗ | 3.6K | 10.9/2.4 | en | 2022 |
| A-OKVQA (Schwenk et al., 2022) | common/factoid | never change | 2-hop | ✗ | ✓ | 24.9K | 8.8/1.3 | en | 2022 |
| InfoSeek (Chen et al., 2023) | fixed kb | never change | 2-hop | ✗ | ✗ | 1.35M | 9.0/1.6 | en | 2023 |
| **Dyn-VQA** | real world | fast/slow/never change | $>$ 2-hop | ✓ | ✓ | 1.5K | 12.5/4.3 | zh/en | 2024 |

**Dataset Statistics** Dyn-VQA is the first dataset specifically proposed for assessing the efficacy of mRAG systems. As shown in Table 2, it contains ~1.5K questions in 9 domains, covering 3 types of question requiring complex dynamic retrieval: questions with rapidly changing answers, questions requiring multi-modal knowledge, multi-hop questions.

Comparative analysis between Dyn-VQA and other knowledge-seeking VQA datasets is also presented in Table 1. It is evident that while other datasets also emphasize the necessity of external knowledge in the question solving, their knowledge typically pertains to only one category. In contrast, each question in Dyn-VQA originates from the real world, encompassing a broader and more heterogeneous range of

Table 2: Statistics of Dyn-VQA.

| Statistic | Number |
|---|---|
| Total Questions | 1452 |
| Domain | 9 |
| English questions | 715 (49.2%) |
| Chinese questions | 737 (50.8%) |
| Questions with fast updating answers | 385 (26.5%) |
| && requiring > 2-hop reasoning | 112 (7.7%) |
| && requiring external visual knowledge | 178 (12.3%) |
| Questions with slow updating answers | 494 (34.0%) |
| Questions with never updating answers | 573 (39.5%) |
| Questions requiring > 2-hop reasoning | 387 (26.7%) |
| && requiring external visual knowledge | 237 (16.3%) |
| Questions requiring ≤ 2-hop reasoning | 1065 (73.3%) |
| Questions requiring external visual knowledge | 865 (59.6%) |
| Questions not requiring external visual knowledge | 587 (40.4%) |
| Average question length | 12.5 |
| Max question length | 60 |
| Average answer length | 4.3 |
| Max answer length | 73 |

knowledge types, and featuring more open-ended answer styles. Reflecting in the length of questions and answers, Dyn-VQA exhibits the longest entries compared to other datasets. Furthermore, Dyn-VQA employs a more systematic question categorization schema, ensuring its challenging.

Table 3: Human performances on different VQA datasets.

| Dataset | Search Count | | | Performance |
| --- | --- | --- | --- | --- |
| | Web Search | Image Search with I. I. | Image Search with T. Q. | |
| VQAv2 | 0.05 | 0 | 0 | 74.31 |
| A-OKVQA | 0.18 | 0.06 | 0 | 60.19 |
| InfoSeek | 0.87 | 0.75 | 0 | 63.67 |
| Dyn-VQA | 1.57 | 0.89 | 0.65 | 55.12 |

Unlike other datasets, which are primarily constructed through templated and automated processes, Dyn-VQA is meticulously curated by humans and requires ongoing human input to maintain the dataset with dynamically updated answers. Consequently, while Dyn-VQA may not match other datasets in scale, it far surpasses them in terms of quality, difficulty, and the cost of each instance. **More details of Dyn-VQA can be found in Appendix.**

**Dataset Difficulty** Questions in Dyn-VQA require more complex external knowledge, whose retrieval process is not fixed. The inherent dynamism of Dyn-VQA naturally ensures its difficulty. To illustrate more intuitively, we also conducted a quantitative comparison of human performance on different datasets. As shown in Table 3, the questions in existing VQA datasets can typically be resolved within two reasoning steps, resulting in an average search count of less than 2 per question. Besides, image search with textual query is not performed at all in other datasets, since the question therein do not require additional visual knowledge except the textual description of the object in image. In terms of overall accuracy, humans achieved the lowest performance on Dyn-VQA, further demonstrating the challenges it presents.

# 4 RETRIEVAL BASELINES AND OMNISEARCH

In this section, we establish several heuristic mRAG baselines and develop our OmniSearch to address dynamic questions that require complex retrieval. Retrieval tools in all methods are Google-based, including web search (textual web content retrieval with textual queries), image search with input images, and image search with textual query.

## 4.1 BASELINES

**Single-hop heuristic mRAG baselines.** The first heuristic baseline is to **retrieve images with input images**, which provides similar images alongside descriptive captions. This method augments MLLMs with visual knowledge about the objects depicted in the input images. Similarly, the second heuristic baseline conducts **web search with input textual questions** and provides MLLMs with the top-k retrieved content as supplementary knowledge. It is acknowledged that these two methods may not furnish precise supportive knowledge, since them only leverage partial modality of the input question as search query. Nonetheless, these baselines are still established to explore the benefits from uni/cross-modality retrieval.

**Two-hop heuristic mRAG baselines.** Typically, existing mRAG methods can be generalized into two primary steps: first, converting the visual concepts in input images into text, and then combining the obtained text with input questions as search queries to retrieve related textual knowledge. For the first step, we employed the **caption of the top-1 image retrieved** by the input image and the **output of the image caption model** as textual representations of the input images, respectively. Subsequently, the web search API is utilized to extract relevant knowledge from the Internet. Compared to single-step mRAG methods, two-step mRAG methods can provide knowledge more precisely related to questions. However, they still encounter the limitation of a fixed retrieval process.

**Estimated Upper Bound.** We estimate the upper bound of mRAGs by the annotation of golden query, which simplifies the question to a last-hop question. Depending on whether the question requires external visual knowledge, either web search or image search API is called. We believe that in this simplified setting, the heuristic mRAG method can provide accurate information to MLLMs.

Descriptions of other baselines are available at Appendix.

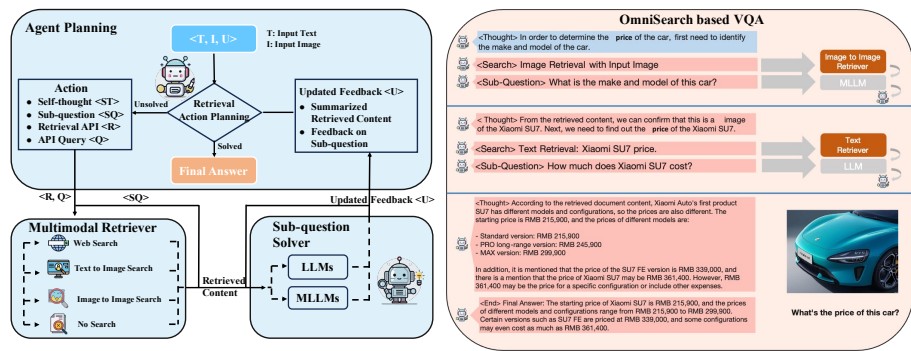

Figure 3: Left: overall framework of OmniSearch. Right: running example of OmniSearch.

## 4.2 OMNISEARCH FRAMEWORK

Heuristic mRAG encounter the issues posed by rigid retrieval processes. A more flexible model that can plan the retrieval strategy tailored to the specific question and dynamically adjusts it throughout the process is desired. Therefore, we propose a self-routing framework called OmniSearch for retrieval planning. The underlying idea behind is to mimic the human of incrementally decomposing the complex question into a sequence of solution actions. As illustrated in Figure 3, overall framework consists of three module. The planning agent is the core module that formulates sub-questions and plans the subsequent retrieval action based on real-world feedback (i.e., retrieved content or solver output). Actual retrieval action is execute by the retriever. Then, sub-question solver generates feedback on sub-question based on the retrieval content and update it to the planner.

**Planning Agent**. Each planned action comprises four critical part: self-thought <ST>, sub-question <SQ>, retrieval API <R>, and API query <Q>. In each step, planning agent comprehends the given question and the real-world feedback in self-thought, then carefully determines the follow-up sub-question to tackle. Meanwhile, different retrieval APIs with query are invoked, depending on the knowledge type required for the sub-question. In a manner akin to human cognitive processes during question-solving, planning agent autonomously generates various potential actions, including: posing additional question to clarify ambiguous or conflicting parts of the retrieved content; refining the retrieval query to acquire more supplementary knowledge; modifying the phrasing of sub-questions; verifying the response to the current sub-question; presenting the next sub-questions; concluding the final answer, etc.

**Retriever** executes actual retrieval operations. In our experiments, web search, image search with text and image retrieval with images are included. More retrieval tools can be considered in future.

**Sub-question Solver** summarizes the retrieved content and endeavors to address the sub-question accordingly. The feedback generated by the solver is then provided to the agent. Notably, the solver is compatible with arbitrary MLLMs or even the planning agent itself, i.e., directly returning the retrieved information to the planner. Depending on computational resources, MLLMs with larger or smaller sizes can be employed. Such pluggable and scalable feature is desired in industrial scenarios, which facilitates the flexible control of GPU cost.

The above process is fully automatic, with OmniSearch performing iterative actions until it believes it has retrieved sufficient knowledge to output a final answer. We trained two versions of OmniSearch based on different MLLMs: proprietary GPT-4V and open-source Qwen-VL-Chat. For GPT-4V, we employed prompt engineering to stimulate its dynamic planning and decision-making capabilities. To facilitate Qwen-VL-Chat's ability to plan and utilize retrieval APIs, we constructed a retrieval API training dataset, leveraging the GPT-4V synthetic data and the existing Infoseek dataset. We train the MLLM in a multi-round conversation mode. Additionally, general instruction data is also utilized to keep the general conversation capability of trained Agent.

The proposal of OmniSearch is inspired by Chain-of-Thought (CoT) Wei et al. (2022), but differs from it essentially. The fundamental distinction between OmniSearch, as a multimodal agent, and CoT is its ability to utilize tools, interact with the environment, and response to the environment Zhang et al. (2023). In contrast, CoT methods primarily stimulate the model's inherent logical

reasoning capabilities through prompts. The CoT approach is unable to decouple intermediate processes, therefore can not be integrated with retrieval tools.

# 5 EXPERIMENTS

## 5.1 EXPERIMENTAL SETTINGS

**Backbone MLLMs for heuristic mRAGs.** We select several advanced MLLMs as backbone model to show their effectiveness equipped with heuristic mRAGs. **Qwen-VL-7B-Chat** is a large visual language model with strong visual and text recognition abilities proposed by Bai et al. (2023b). **GPT-4V** and **Qwen-VL-Max**[2] are also selected as representatives of the closed-source models to show how the larger MLLMs will affect the results. Additionally, **Qwen-7B-Chat** is included in our experiments, which is a text-only LLM. We use this model to assess how multimodal RAG can solve the visual problem for textual LLM.

**Evaluation Metric.** The automated metric *F1-Recall* is utilized as the evaluation metric. We calculate the ratio of common tokens between model-generated responses and ground truth. Specifically, we first segment the generated text and golden text into token lists using word segmentation tools[3], then calculate the ratio of tokens generated by models belonging to the golden token list.

## 5.2 MAIN RESULTS

Table 4: Main results on Dyn-VQA.

| Model | Answer Update Frequency | | | Reasoning Steps | | Visual-Seeking | | Language | | all |
|---|---|---|---|---|---|---|---|---|---|---|
| | fast | slow | never | ≤ 2-hop | > 2-hop | no | yes | zh | en | |
| *Original (M)LLMs* | | | | | | | | | | |
| Qwen-VL-Chat | 13.69 | 14.20 | 17.27 | 15.56 | 14.49 | 15.50 | 15.13 | 16.92 | 13.58 | 15.28 |
| Qwen-7B-Chat | 5.63 | 7.86 | 15.97 | 10.48 | 10.43 | 11.05 | 10.08 | 10.48 | 10.46 | 10.47 |
| Qwen-VL-Max | 15.11 | 30.44 | 39.51 | 30.22 | 29.21 | 31.10 | 29.18 | 22.81 | 37.32 | 29.96 |
| GPT-4V | 17.63 | 27.80 | 40.82 | 30.80 | 28.74 | 31.71 | 29.26 | 26.44 | 34.18 | 30.25 |
| *+ Heuristic mRAG: Retrieving Images with Input Images* | | | | | | | | | | |
| Qwen-VL-Chat | 15.74 | 17.12 | 25.74 | 20.52 | 19.14 | 22.68 | 18.44 | 22.06 | 18.19 | 20.16 |
| Qwen-7B-Chat | 10.97 | 15.04 | 25.91 | 18.76 | 16.86 | 24.18 | 14.23 | 16.80 | 19.75 | 18.25 |
| Qwen-VL-Max | 24.04 | 28.99 | 45.49 | 34.22 | 34.08 | 41.54 | 29.19 | 31.07 | 37.39 | 34.19 |
| GPT-4V | 20.18 | 33.21 | 50.00 | 35.94 | 35.65 | 42.63 | 31.33 | 30.90 | 40.32 | 35.87 |
| *+ Heuristic mRAG: Retrieving Web Pages with Input Questions* | | | | | | | | | | |
| Qwen-VL-Chat | 20.78 | 18.27 | 27.61 | 22.76 | 22.20 | 23.34 | 22.07 | 26.66 | 17.94 | 22.59 |
| Qwen-7B-Chat | 14.65 | 15.47 | 24.98 | 19.14 | 18.66 | 19.99 | 18.34 | 17.93 | 20.12 | 19.01 |
| Qwen-VL-Max | 26.71 | 27.37 | 35.84 | 30.65 | 30.22 | 31.14 | 30.13 | 30.27 | 30.82 | 30.54 |
| GPT-4V | 22.48 | 30.92 | 40.84 | 33.00 | 31.47 | 34.32 | 31.42 | 31.10 | 34.13 | 32.59 |
| *+ Heuristic Two-Step mRAG: Retrieving Image First, then Retrieving Web Pages with Question appended to Retrieved Caption* | | | | | | | | | | |
| Qwen-VL-Chat | 19.17 | 20.02 | 28.54 | 23.33 | 22.68 | 23.84 | 22.69 | 24.11 | 22.17 | 23.16 |
| Qwen-7B-Chat | 15.27 | 17.33 | 28.53 | 21.70 | 19.83 | 26.65 | 17.50 | 20.47 | 21.96 | 21.20 |
| Qwen-VL-Max | 24.44 | 30.75 | 43.21 | 34.03 | 33.91 | 38.26 | 31.10 | 32.04 | 36.01 | 33.99 |
| GPT-4V | 20.37 | 33.98 | 48.46 | 36.12 | 36.04 | 40.19 | 33.32 | 32.99 | 39.30 | 36.10 |
| *+ Heuristic Two-Step mRAG: Image Caption First, then Retrieving Web Pages with Question appended to Caption* | | | | | | | | | | |
| Qwen-VL-Chat | 22.05 | 25.87 | 31.84 | 27.58 | 26.21 | 27.44 | 27.06 | 28.81 | 25.57 | 27.21 |
| Qwen-7B-Chat | 14.65 | 21.16 | 28.66 | 22.89 | 21.02 | 23.64 | 21.55 | 16.57 | 28.39 | 22.39 |
| Qwen-VL-Max | 24.27 | 32.93 | 44.03 | 35.04 | 34.97 | 35.16 | 34.92 | 31.10 | 39.05 | 35.02 |
| GPT-4V | 24.90 | 36.74 | 45.76 | 37.23 | 36.94 | 37.82 | 36.70 | 31.65 | 42.81 | 37.15 |
| *Generative Search Engine* | | | | | | | | | | |
| Bing Chat | 27.71 | 32.77 | 32.99 | 31.67 | 30.80 | 35.44 | 28.64 | 29.62 | 32.74 | 31.44 |
| Perplexity AI | 29.62 | 34.69 | 34.88 | 33.67 | 32.81 | 37.46 | 30.67 | 31.59 | 34.80 | 33.51 |
| Gemini | 36.17 | 32.86 | 42.84 | 38.75 | 34.78 | 46.39 | 31.82 | 35.77 | 39.69 | 37.69 |
| *Ours* | | | | | | | | | | |
| OmniSearch (Qwen-VL-Chat) | 35.16 | 40.89 | 45.52 | 41.34 | 40.81 | 42.56 | 40.28 | 39.22 | 43.23 | 41.20 |
| OmniSearch (GPT-4V) | **44.04** | **49.58** | **54.45** | **50.38** | **49.06** | **50.49** | **49.73** | **46.96** | **53.21** | **50.03** |
| *Estimated Upper Bound: + Retrieving Web Pages with Gloden Query* | | | | | | | | | | |
| Qwen-VL-Chat | 37.46 | 46.52 | 52.18 | 46.73 | 45.28 | 47.94 | 45.27 | 43.88 | 48.90 | 46.35 |
| Qwen-7B-Chat | 39.69 | 47.27 | 57.76 | 49.53 | 49.02 | 50.94 | 48.36 | 46.02 | 52.88 | 49.40 |
| Qwen-VL-Max | 42.19 | 53.01 | 56.60 | 51.91 | 50.58 | 52.97 | 50.60 | 49.83 | 53.33 | 51.56 |
| GPT-4V | 45.59 | 54.23 | 60.78 | 55.15 | 52.81 | 54.53 | 54.51 | 51.08 | 58.07 | 54.52 |
| Human Performance | 51.63 | 60.02 | 53.19 | 54.12 | 58.31 | 57.86 | 53.20 | 51.96 | 58.36 | 55.12 |

The performance of various MLLMs with different mRAG methods are shown in Table 4, from which we can find that:

---

[2]Our evaluation was conducted up to July 1, 2024.

[3]Jieba (https://github.com/fxsjy/jieba) for Chinese, and NLTK (https://www.nltk.org/) for English.

(1) Our OmniSearch (GPT-4V) significantly outperforms other models, encompassing both state-of-the-art MLLMs with heuristic mRAGs and commercial generative search engines. Even Qwen-VL-Chat-based OmniSearch surpasses the considerably larger GPT-4V equipped with two-step heuristic mRAG. We attribute this to two aspects: on the one hand, the OmniSearch decomposes a complex question into a sequence of sub-questions, reducing the retrieval burden in a single step. On the other hand, it rethinks the retrieved content and sub-questions to ensure the accuracy of the sub-answers, mitigating the risk of error propagation.

(2) Regarding overall performance, the OmniSearch closely parallel human and GPT-4V enhanced with content retrieved via gold query, highlighting its superior abilities. Nevertheless, a significant gap remains between the OmniSearch and human performance on questions belong to the three most challenging subcategories (fast-changing, >2-hop, requiring external visual knowledge), which indicates substantial room for improvement in agent-based mRAG for real-world questions. How to generate more human-like search logic is a promising direction for future research.

(3) Despite achieving more than 50% performance, Dyn-VQA remains a formidable challenge for both AI systems and humans. It is observed that for questions necessitating multi-step retrieval or additional visual knowledge, all models consistently underperform compared to other questions within the same classification schema. Especially for questions with different answer update frequencies, the variance in model performance is high. We can conclude that questions requiring rapidly changing knowledge pose the most intractable challenge, as such knowledge cannot be internalized by MLLMs.

(4) For two-step heuristic mRAGs, leveraging image caption model to transform visual concepts brings more gain to the original MLLMs, which provides a more detailed image description for the next retrieval step. However, this advantage reverses for questions that do not require additional visual knowledge, primarily because the majority of them are 2-hop (74%) and do not demand visual knowledge beyond the concepts presented in the image itself. Supplementary information from the image caption model does not substantially benefit the model.

(5) Commercial generative search engines generally perform poorly on Dyn-VQA. Even the best-performing engine, Gemini, only matching the performance of GPT-4V with two-step mRAG. Further case analysis reveals these generative search engines lack essential grounding capabilities: they fail to associate "it" in the question with objects in the image, nor can integrate multimodal information effectively. This suggests that questions in Dyn-VQA represent the real demand in industrial scenarios.

(6) Comparing Qwen-7B-Chat and Qwen-VL-Chat, we observe that the performance gap between the models is reduced once equipped with mRAG. This phenomenon indicates that mRAG can assist pure text LLMs in addressing multi-modal issues.

Due to space constraints, more analysis experiments are placed **in the appendices in the Supplementary Material, and they are highly recommended to the reader**.

Table 5: Experiments on OmniSearch paired with different MLLMs as sub-question solvers. OmniSearch (G) and OmniSearch (Q) refer to OmniSearch implementations based on GPT-4V and Qwen-VL-Chat, respectively.

| Planning Model | Sub-question Solver | Answer Update Frequency | | | Reasoning Steps | | Visual-Seeking | | Language | | all |
|---|---|---|---|---|---|---|---|---|---|---|---|
| | | fast | slow | never | ≤ 2-hop | > 2-hop | no | yes | zh | en | |
| OmniSearch (Q) | OmniSearch (Q) | 35.16 | 40.89 | 45.52 | 41.34 | 40.81 | 42.56 | 40.28 | 39.22 | 43.23 | 41.20 |
| OmniSearch (Q) | GPT-4V | 37.14 | 42.82 | 47.48 | 43.29 | 42.78 | 44.46 | 42.26 | 41.21 | 45.15 | 43.15 |
| OmniSearch (Q) | GPT-4V + GPT-4 | 38.98 | 44.52 | 49.18 | 45.03 | 44.52 | 46.20 | 44.00 | 42.97 | 46.87 | 44.89 |
| OmniSearch (Q) | Qwen-VL-Chat | 34.10 | 39.88 | 44.50 | 40.32 | 39.77 | 41.53 | 39.25 | 38.18 | 42.22 | 40.17 |
| OmniSearch (G) | OmniSearch (G) | 44.04 | 49.58 | 54.45 | 50.38 | 49.06 | 50.49 | 49.73 | 46.96 | 53.21 | 50.03 |
| OmniSearch (G) | Qwen-VL-Chat | 38.65 | 44.68 | 52.25 | 46.56 | 44.72 | 49.40 | 43.80 | 41.63 | 50.64 | 46.07 |

## 5.3 ANALYSIS EXPERIMENTS ON OMNISEARCH

In this section, we conduct extensive analysis experiments to answer the following questions on our OmniSearch:

**How different models as sub-question solvers affect overall performance?** As shown in Table 5, several observations can be made regarding the performance of OmniSearch when paired with different MLLMs as sub-question solvers:

Table 6: Comparison of token costs and expenses for different models.

| Planning Model | Sub-question Solver | # Input Tokens | # Output Tokens | Expenses ($\times 10^{-3}$\$) | Performance |
|---|---|---|---|---|---|
| Two-Step mRAG | GPT-4V | 1454.0 (G) | 132.5 (G) | 18.5 | 37.15 |
| Two-Step mRAG | Qwen-VL-Chat | 749.9 (Q) | 28.6 (Q) | 0.2 | 27.21 |
| OmniSearch (G) | OmniSearch (G) | 3028.5 (G) | 476.9 (G) | 44.6 | 50.03 |
| OmniSearch (G) | Qwen-VL-Chat | 1217.2 (G) + 2073.4 (Q) | 386.0 (G) + 124.8 (Q) | 24.4 | 46.07 |
| OmniSearch (Q) | OmniSearch (Q) | 9578.3 (Q) | 572.5 (Q) | 3.2 | 41.20 |
| OmniSearch (Q) | GPT-4V | 2371.5 (G) + 992.2 (Q) | 281.0 (G) + 551.4 (Q) | 32.8 | 43.15 |

(1) In the case of the Qwen-VL-Chat Based OmniSearch, employing the larger model GPT-4V as the sub-question solver significantly enhances the performance of the OmniSearch, indicating the continued validity of scaling laws for sub-question solver. Meanwhile, substituting the sub-question solver of the GPT-4V Based OmniSearch with the smaller Qwen-VL-Chat leads to a predictable decrease in model performance. Nonetheless, it still outperforms Qwen-VL-Chat with two-step heuristics mRAG from Table 4.

(2) we also explored a more complex invocation strategy for the sub-question solving model: leveraging GPT-4V for sub-questions entailing multimodal contexts, and employing GPT-4 for those involving purely textual contexts, which is considered to be more capable on text-only questions compared to GPT-4V. This approach contributes further to performance enhancement. More refined invocation strategies is also worthy to be explored, such as having the sub-question solver output bounding boxes for certain objects in the image to guide more precise retrieval. We leave this topic in the future work.

(3) To assess whether the OmniSearch with retrieval path planning learning has been impaired in its question solving ability, we replaced the sub-question solver of the Qwen-VL-Chat Based OmniSearch with the original Qwen-VL-Chat. Comparison of rows 1 and 4 in Table 5 reveals that employing the OmniSearch as the sub-question solver instead improves question-solving ability. This enhancement demonstrate that the learning of retrieval path planning also involves the ability to understand and reason about retrieval knowledge, potentially enhancing the model's problem solving ability and yielding cross-task gains.

**How different models as sub-question solvers affect token and expenses?** In Table 6, we further examine the token costs and actual expenses[4] brought by different sub-question solvers. Although more costly, the enhancement provided by the OmniSearch relative to the heuristic mRAG is considerably substantial. The correlation between the performance of the OmniSearch and the actual expenses is proportional, yet not linear. When substituting GPT-4V with Qwen-VL-Chat as the sub-question solver (rows 3 and 4), the absolute performance declines by under 4 points, approximately 7.9%, while the expenditure is nearly halved, demonstrating the excellent scalability of OmniSearch. The results also indicate that sub-question reasoning is not the bottleneck of current methods, rather the retrieval strategy of complex questions presents a more urgent challenge. This conclusion is further supported by comparing the enhancements achieved by substituting the planning model (row 6 to 5) and sub-question solver (row 6 to 7) of the OmniSearch (Q) with GPT-4V. The benefits realized by the former are more pronounced. Consequently, when computational resources are constrained, priority should be given to ensuring that retrieval planning model can utilize a larger model as backbone.

## 6 ANALYSIS EXPERIMENTS ON DYN-VQA DATASET

### 6.1 PERFORMANCE COMPARISON ON DIFFERENT DOMAINS

Figure 4 illustrates the performance of Qwen-VL-Chat and GPT-4V equipped with different mRAG methods across various domains. We can intuitively observe that each mRAG method enhances the efficacy of the original model. The coverage of both original models is notably broadened by the mRAG methods, particularly for the smaller Qwen-VL-Chat. However, in several domains, such as transportation, the Qwen-VL-Chat-based OmniSearch instead exhibits superior performance

---

[4]We refer https://azure.microsoft.com/zh-cn/pricing/details/cognitive-services/openai-service/ for GPT-4V price.

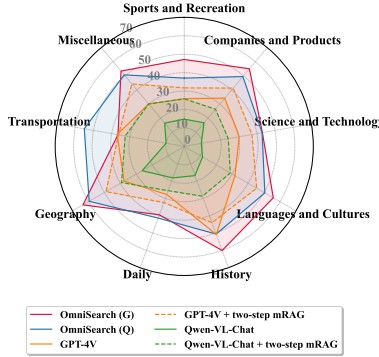 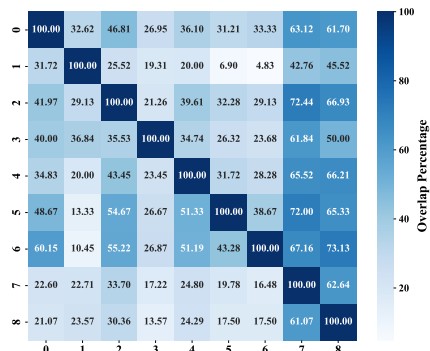

Figure 4: Model performance of different domains.

Figure 5: Pairwise overlap between correctly answered questions of different models. 0-8: Qwen-VL-Chat, Qwen-7B-Chat, Deepseek-VL-7B-Chat, VisualGLM-6B, Llava-V1.6-Mistral-7B, mPLUG-Owl2.1, InstructBLIP-Vicuna-7B, Qwen-VL-Max, GPT-4V.

compared to the GPT-4V-based one. Further analysis reveals that this phenomenon is primarily attributed to the long-tail property of transportation domain, which contains only 10 VQA instances, with the majority comprising 2-hop questions or questions that do not involve changing knowledge. In these cases, GPT-4V-based OmniSearch tends to over-retrieve, e.g., it has retrieved the necessary information but over-cautiously continues to gather additional information to validate the answer, resulting in the correct answer being obscured within a large volume of retrieved information. This underscores the need for ongoing enhancements to the robustness of our OmniSearch.

## 6.2 PREDICTION OVERLAP

In this subsection, we investigate the overlap of questions correctly answered by different models. Firstly, we observed that no questions in the Dyn-VQA were correctly answered by all models, and 31% of the questions did not receive a correct prediction from any model. Figure 5 illustrates the degree of pairwise overlap in correctly answering questions on Dyn-VQA. Each row indicates the proportion of questions correctly answered by the corresponding model that were also correctly answered by other models. Overall, the two highest-performing models, Qwen-VL-Max and GPT-4V, exhibited relatively high overlap, but still hover around 60%. Furthermore, looking at heat blocks (6, 8), we find that even for InstructBLIP-Vicuna-7B, which demonstrated the weakest performance (12.33 overall F1 recall, as detailed in Table 11 in the Appendix), 26.87% (100 - 73.13) of the questions it successfully answered could not be correctly answered by the best-performing GPT-4V. This indicates substantial differences in model behavior and shows that although some models generally outperform others, their superiority is not attributable to correctly answering the "hard" questions while consistently getting the "easy" ones right. The varied challenges presented by Dyn-VQA affect models differently, highlighting ensemble-based and self-consist-based approaches as promising directions for future research.

## 7 CONCLUSION

In this paper, we study the multimodal retrieval augmented generation (mRAG). We argue that existing heuristic mRAGs typically predefined fixed retrieval processes, which causes two issues: (1) Non-adaptive Retrieval Queries. (2) Overloaded Retrieval Queries. However, these rigidity issues cannot be adequately reflected by current knowledge-seeking visual question answering (VQA) datasets. Therefore, we first construct Dyn-VQA dataset, consisting of three types of "dynamic" questions, which require complex knowledge retrieval strategies variable in query, tool, and time. Furthermore, OmniSearch is proposed as the first self-adaptive planning agent for multimodal retrieval. Extensive experiments prove the effectiveness of our OmniSearch, also highlight the challenges posed by Dyn-VQA.

## ACKNOWLEDGEMENTS

This research project is supported by National Natural Science Foundation of China (Grant No.62276154), Research Center for ComputerNetwork (Shenzhen) Ministry of Education, the Natural Science Foundation of Guangdong Province (Grant No.2023A1515012914 and 440300241033100801770), Basic Research Fund of Shenzhen City (Grant No.JCYJ20210324120012033, JCYJ20240813112009013 and GJHZ20240218113603006), the Major Key Projectof PCL for Experiments and Applications (PCL2022A05 and PCL2023A09), NSF under Grants III-2106758, and POSE-2346158.

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

## A    MORE DETAILS ON DYN-VQA DATASET

### A.1    STATISTICS

Figure 6 illustrates the data distribution across various domains and the answer change frequencies in Dyn-VQA. Among the 9 domains, Sports and Recreation, and Companies and Products constitute approximately 50% of the data. The distribution of questions with fast, slow, and never answer changes is relatively balanced within the classes and does not exhibit a long tail, reflecting a distribution that closely aligns with real-world scenarios.

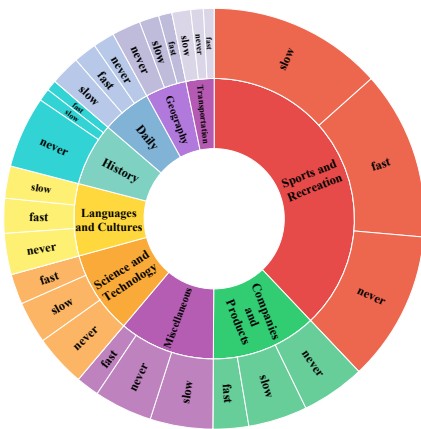

Figure 6: Data distribution of different domains and answer change frequencies on the Dyn-VQA dataset.

### A.2    DATASET QUALITY

To ensure dataset quality, following the initial annotation of Dyn-VQA, we employed two quality control (QC) annotators to re-evaluate the dataset. This re-evaluation included verification of the answers, domains, answer update frequencies, reasoning steps, and whether require external visual knowledge for each data instance. Data deemed incorrect by both QC annotators was filtered out. The agreement between QC annotators #1 and #2 with the initial annotations, as well as the agreement between the two QC annotators, are presented in Table 7. The inter-annotator agreement measured by Fleiss's Kappa (Fleiss, 1971) all exceeded 0.8, demonstrating the reliability of the annotation results.

Table 7: The inter-annotator agreement of QC#1, QC#2 and initial annotation with each other.

|  | Answer | Domain | Answer Update Freq. | Reasoning Step | External Visual-Seek |
|---|---|---|---|---|---|
| Init. vs. QC#1 | 81.2 | 84.4 | 89.6 | 84.1 | 87.9 |
| Init. vs. QC#2 | 84.2 | 85.6 | 85.4 | 82.3 | 84.5 |
| QC#1 vs. QC#2 | 80.1 | 83.8 | 86.9 | 81.7 | 85.0 |
| Avg. | 81.8 | 84.6 | 87.3 | 82.7 | 85.8 |

Table 8: Question and answer diversity. Mean pairwise cosine distances are used as metrics.

| Dataset | Question | Answer |
|---|---|---|
| VQAv2 | 0.8405 | 0.7606 |
| A-OKVQA | 0.8428 | 0.9078 |
| InfoSeek | 0.7569 | 0.8918 |
| Dyn-VQA | 0.8532 | 0.9135 |

### A.3    DATASET DIVERSITY

To assess the diversity of Dyn-VQA compared with other datasets, we calculated pairwise cosine distances for each dataset. Following A-OKVQA, we utilized a sentence-transformers[5] as the encoder. As indicated in Table 8, Dyn-VQA exhibits more diverse data types, evidenced by larger cosine distances. Intuitively, questions in InfoSeek are primarily constructed through templates, resulting

---

[5]https://huggingface.co/sentence-transformers/multi-qa-MiniLM-L6-cos-v1

in more homogeneous data format. In contrast, Dyn-VQA is manually curated. Moreover, since the questions of Dyn-VQA are all from open-ended domains, the answers of them feature longer response length compared with the same manually constructed VQAv2 and A-OKVQA, which can essentially be answered with a single word.

## A.4 Regular Updates to Dyn-VQA

The biggest characteristic of Dyn-VQA is that the knowledge required to answer the questions it contains is dynamically updated over time, that is, the answers to the questions in Dyn-VQA are constantly changing. Therefore, to ensure that Dyn-VQA can serve as an effective research resource for the community in the long term, it is necessary for us to dynamically update and maintain Dyn-VQA's answer annotation information. **In general, based on the analysis of the frequency of change of Dyn-VQA's answer, we commit to updating Dyn-VQA's answer annotation every three months to ensure its timeliness.**

In particular, we plan to implement a semi-automatic data update mechanism in which models and humans work together. Specifically, For a certain sample in Dyn-VQA, we first retrieve relevant text knowledge through the search engine, and then we use LLMs such as Qwen1.5-72B to compare the latest knowledge retrieved with the original answer to determine whether the original answer for the sample needs to be updated. Note that here we only need LLMs to determine whether the answer needs to be updated, and do not need them to accurately answer the latest answer, because we think this is simpler and more friendly for LLMs. After we have the prediction results of LLMs as a basis, we then carry out the manual update process. We require every human annotator involved in the data update process to accurately update each sample's answer based on their common sense, the results of large model judgments, and the latest relevant knowledge retrieved by the search engines. We believe that the semi-automatic update mechanism, in which models and humans cooperate, will not only reduce the workload of manual annotators but also improve the accuracy of data updates.

## B Related Works

### B.1 Multimodal Large Language Models

In 2023, with the advent of GPT-4V (Achiam et al., 2023), a series of MLLMs (Bai et al., 2023b; Lu et al., 2024a; Dai et al., 2023; Liu et al., 2024c; Wang et al., 2023) have been proposed and demonstrated superior results on a variety of vision-language tasks (Yin et al., 2023; Wu et al., 2023). Despite promising results, MLLMs tend to haphazardly produce responses that appear plausible yet contain factual errors when faced with real-world questions. Many prior works explore mitigating this hallucination issue (Liu et al., 2024a; Bai et al., 2024) by introducing additional knowledge-enhancing data or tasks into the different training stages of MLLMs, including pre-training (Zhai et al., 2023), instruction fine-tuning (Chai et al., 2024b; Jain et al., 2024), and RLHF (Sun et al., 2023; Yu et al., 2024). However, the expensive training cost of MLLMs pose significant challenges to the scalability of these methods. Therefore, mRAG attracts growing interest as an effective and efficient alternative.

### B.2 Multimodal Retrieval Augmented Generation

Besides retrieval from knowledge bases as described in Related Works section in main content, some work explores retrieval from other knowledge sources. REVEAL (Hu et al., 2023) integrates knowledge retrieved from multiple sources, including wikidata, wikipedia and other VQA datasets. PICa (Yang et al., 2022) consider LLMs as implicit knowledge bases and extract relevant information from GPT-3 (Brown et al., 2020) with image description as prompt.

### B.3 VQA datasets as mRAG Benchmarks

Knowledge-seeking VQA datasets (Marino et al., 2019; Jain et al., 2021; Schwenk et al., 2022; Kil et al., 2024; Chen et al., 2024b) are widely employed to evaluate the performance of mRAGs, which rely on external information to address open-domain visual questions. For instance, the recently introduced Wikipedia-based VQA dataset, InfoSeek (Chen et al., 2023), emphasizes fine-

grained entity knowledge for open-domain questions. A-OKVQA (Schwenk et al., 2022) is a new knowledge-based VQA benchmark that necessitates a broad spectrum of commonsense and world knowledge. As illustrated in Table 1, several knowledge-seeking VQA datasets have been proposed in recent years.

Nevertheless, the knowledge scope assessed by these datasets is constrained. More critically, the questions in these datasets often exhibit a fixed format, typically querying a specific property of the object in image, which is generally textual knowledge available on the internet. Such two-hop questions can be readily addressed by standard two-step retrieval. The static nature of these datasets prompts us to propose Dyn-VQA, which requires the retrieval of dynamic knowledge, knowledge from a more diverse range of modalities, or more complex multi-hop knowledge.

## C    EXPERIMENTS

### C.1    BASELINES

In the main experiment of Table 4, we also introduced generative search engine and human as baseline models. We describe them following:

**Generative Search Engine** Among the commercial AI products, LLM-powered generative search engines like Bing Chat, PerplexityAI, mita.ai, Tongyi, and GPT-4o stand out. For our experiment, we select Bing Chat, pro-version of PerplexityAI, and Gemini-Advance as representatives of AI search engines. They have multimodal RAG ability for fair comparison.

**Human Performance.** We also investigated human performance on Dyn-VQA, employing participants with at least a bachelor degree. These participants were not involved in the Dyn-VQA annotation process.

### C.2    OTHER BACKBONE MLLMS FOR HEURISTIC MRAGS

In the Appendix, we supplement the main experiment with more MLLM as backbone for heuristic mRAGs. **Deepseek-VL-7B-Chat** is an open-source large visual language model introduced by Lu et al. (2024b). **VisualGLM-6B** is an open-source, bilingual, multi-modal large visual language model proposed by Du et al. (2022). **Llava-V1.6-Mistral-7B** is an open-source large visual language model proposed by Liu et al. (2024b), which is trained by fine-tuning LLM on multimodal instruction-following data based on Jiang et al. (2023). **mPLUG-Owl2.1** (Ye et al., 2024) is a large visual language model trained with a two-stage method for aligning image and text.

### C.3    TRAINING DETAILS

OmniSearch (G) is constructed by prompt engineering for GPT-4V, whose prompt template can be found in the next subsection. OmniSearch (Q) is developed by instruction fine-tuning of Qwen-VL-Chat. we synthesize instruction data containing planning trajectories by using GPT-4V and raw InfoSeek data. ∼40K data is synthesized, and then the data is filtered by a sequence of predefined rules. 13K data is eventually obtained. Additionally, general instruction data from CogVLM-SFT-311K[6] is mixed in a ratio of 1:2 with the planning instruction data. We use LoRA to parameter-efficient fine-tune Qwen-VL-Chat, with LoRA rank and alpha are 8 and 32. AdamW (Loshchilov & Hutter, 2019) optimizer is employed for model training, with learning rate set of 1e-4 and weight decay of 0.1. We utilize a cosine learning rate schedule, warming up over 5% of the training steps. The model is fine-tuned with 1 epochs, with the batch size per device set to 4 and the gradient accumulation step set to 8. The maximum sequence length is 8192. The training are run on 4 NVIDIA A100 SXM4 80GB GPUs.

All data is in the form of multi-round conversations. During model training, we expect the model to learn to generate response given the instruction and input text, thus we compute the loss function by considering only the response tokens of each round and ignoring the input tokens.

---

[6]https://huggingface.co/datasets/THUDM/CogVLM-SFT-311K

Table 9: Experiments on the impact of using different parts of retrieved content. We report the performance of OmniSearch (G).

| Retrieved Content | Using Part | Answer Update Frequency | | | Reasoning Steps | | Visual-Seeking | | Language | | all |
|---|---|---|---|---|---|---|---|---|---|---|---|
| | | fast | slow | never | ≤ 2-hop | > 2-hop | no | yes | zh | en | |
| Image | Image & Caption | 44.04 | 49.58 | 54.45 | 50.38 | 49.06 | 50.49 | 49.73 | 46.96 | 53.21 | 50.03 |
| | - Caption | 41.65 | 47.23 | 52.03 | 48.00 | 46.67 | 48.02 | 47.39 | 44.57 | 50.81 | 47.64 |
| | - Image | 42.58 | 48.14 | 52.97 | 48.93 | 47.60 | 48.98 | 48.30 | 45.50 | 51.74 | 48.57 |
| Web Snippet | Web Title & Description | 44.04 | 49.58 | 54.45 | 50.38 | 49.06 | 50.49 | 49.73 | 46.96 | 53.21 | 50.03 |
| | + Related Knowledge | 45.26 | 50.78 | 55.68 | 51.60 | 50.28 | 51.75 | 50.92 | 48.17 | 54.43 | 51.25 |
| | - Web Title | 42.09 | 47.66 | 52.47 | 48.44 | 47.11 | 48.47 | 47.82 | 45.01 | 51.25 | 48.08 |

## C.4 SUPPLEMENT TO MAIN EXPERIMENTS

Experiment results of using more MLLMs as backbones for heuristic mRAGs are supplemented in Table 10, from which we can find:

(1) Among all open-sourced MLLMs, Deepseek-VL-7B-Chat exhibits the best performance as a backbone, achieving the highest overall performance. While heuristic mRAG methods show the most significant improvements with InstructBLIP-Vicuna-7B, where the average absolute gain across four heuristic mRAG approaches reaches 16.60, marking an enhancement of 223.1% compared to the baseline performance of InstructBLIP-Vicuna-7B. This substantial increase is likely attributed to the initially lower performance of the original InstructBLIP-Vicuna-7B.

(2) The performance variance between MLLMs that enhanced by the heuristic mRAG becomes smaller. For instance, with the incorporation of the image caption-based two-step mRAG method, the variance in performance among open-source MLLMs reduces from 4.92 to 2.42. Combined with the analysis in (1), this suggests that for MLLMs with suboptimal foundational capabilities, mRAG serves as an optimal method to bolster model performance. It is not only less resource-dependent, but also convenient to deploy.

(3) Although single-step heuristic mRAGs may not retrieve the most precise content, they still benefit the original model capabilities. Retrieving image information with input images enriches the MLLM with information about the objects in images, whereas web page retrieval with input questions can also retrieve some relevant information due to certain keywords in the questions. Overall, web page retrieval yields a greater improvement with an average of 6.78, which might be attributable to the inherent capacity of MLLMs to recognize some objects depicted in the images. It also indicates that unlike previous VQA datasets, the challenge of Dyn-VQA does not lie solely in the recognition of objects in images.

## D ANALYSIS EXPERIMENTS ON OMNISEARCH

### D.1 ANALYSIS EXPERIMENTS ON RETRIEVED CONTENT

**Is each part of the retrieved content useful?** Table 9 presents the impact of utilizing different parts of the content returned by the retrieval APIs, from which we can find that each part of the retrieved content is beneficial to overall performance. Utilizing different single part results in varying degrees of performance degradation compared to using the entire retrieved content. Notably, image captions contribute most to final performance. This phenomenon arises primarily because nearly all questions in Dyn-VQA necessitate object recognition in images, for which captions from retrieved similar images provide crucial additional information to the model. In contrast, the benefit derived from incorporating relevant knowledge provided by the search engine is relatively modest. This is predominantly due to such information typically constituting static background knowledge that lacks direct relevance to the specific problem at hand.

**Is more retrieved content useful?** Table 11 explores the impact of varying amounts of retrieved content on model performance, from which we can find that:

(1) The model incorporating retrieved content consistently surpasses its counterpart without retrieval, affirming the inherent advantage of mRAG.

Table 10: Main results on Dyn-VQA.

| Model | Answer Update Frequency | | | Reasoning Steps | | Visual-Seeking | | Language | | all |
|---|---|---|---|---|---|---|---|---|---|---|
| | fast | slow | never | ≤ 2-hop | > 2-hop | no | yes | zh | en | |
| *Original (M)LLMs* | | | | | | | | | | |
| Qwen-VL-Chat | 13.69 | 14.20 | 17.27 | 15.56 | 14.49 | 15.50 | 15.13 | 16.92 | 13.58 | 15.28 |
| Qwen-7B-Chat | 5.63 | 7.86 | 15.97 | 10.48 | 10.43 | 11.05 | 10.08 | 10.48 | 10.46 | 10.47 |
| Deepseek-VL-7B-Chat | 11.66 | 18.57 | 30.16 | 22.12 | 19.08 | 23.26 | 19.99 | 19.94 | 22.73 | 21.31 |
| VisualGLM-6B | 12.05 | 13.94 | 20.92 | 16.63 | 14.99 | 18.92 | 14.34 | 16.21 | 16.18 | 16.19 |
| Llava-V1.6-Mistral-7B | 15.39 | 19.72 | 28.34 | 22.65 | 20.12 | 23.53 | 20.92 | 16.27 | 27.85 | 21.97 |
| mPLUG-Owl2.1 | 8.82 | 11.25 | 18.44 | 13.81 | 12.44 | 14.48 | 12.74 | 8.05 | 19.01 | 13.44 |
| InstructBLIP-Vicuna-7B | 6.08 | 7.40 | 8.38 | 7.48 | 7.33 | 7.51 | 7.38 | 4.65 | 10.31 | 7.44 |
| Qwen-VL-Max | 15.11 | 30.44 | 39.51 | 30.22 | 29.21 | 31.10 | 29.18 | 22.81 | 37.32 | 29.96 |
| GPT-4V | 17.63 | 27.80 | 40.82 | 30.80 | 28.74 | 31.71 | 29.26 | 26.44 | 34.18 | 30.25 |
| *+ Heuristic mRAG: Retrieving Images with Input Images* | | | | | | | | | | |
| Qwen-VL-Chat | 15.74 | 17.12 | 25.74 | 20.52 | 19.14 | 22.68 | 18.44 | 22.06 | 18.19 | 20.16 |
| Qwen-7B-Chat | 10.97 | 15.04 | 25.91 | 18.76 | 16.86 | 24.18 | 14.23 | 16.8 | 19.75 | 18.25 |
| Deepseek-VL-7B-Chat | 15.95 | 22.63 | 36.80 | 26.49 | 26.35 | 32.27 | 22.50 | 25.97 | 26.94 | 26.45 |
| VisualGLM-6B | 15.51 | 16.69 | 29.08 | 21.73 | 19.99 | 25.75 | 18.22 | 20.84 | 21.71 | 21.27 |
| Llava-V1.6-Mistral-7B | 17.38 | 19.74 | 33.48 | 25.42 | 22.10 | 32.00 | 19.47 | 17.01 | 32.29 | 24.54 |
| mPLUG-Owl2.1 | 13.49 | 16.52 | 28.14 | 21.42 | 17.20 | 25.84 | 16.54 | 13.45 | 27.36 | 20.30 |
| InstructBLIP-Vicuna-7B | 13.40 | 15.83 | 29.02 | 20.19 | 15.16 | 25.15 | 16.68 | 13.10 | 25.05 | 20.07 |
| Qwen-VL-Max | 24.04 | 28.99 | 45.49 | 34.22 | 34.08 | 41.54 | 29.19 | 31.07 | 37.39 | 34.19 |
| GPT-4V | 20.18 | 33.21 | 50.00 | 35.94 | 35.65 | 42.63 | 31.33 | 30.90 | 40.32 | 35.87 |
| *+ Heuristic mRAG: Retrieving Web Pages with Input Questions* | | | | | | | | | | |
| Qwen-VL-Chat | 20.78 | 18.27 | 27.61 | 22.76 | 22.20 | 23.34 | 22.07 | 26.66 | 17.94 | 22.59 |
| Qwen-7B-Chat | 14.65 | 15.47 | 24.98 | 19.14 | 18.66 | 19.99 | 18.34 | 17.93 | 20.12 | 19.01 |
| Deepseek-VL-7B-Chat | 20.34 | 23.44 | 32.02 | 26.13 | 25.65 | 27.48 | 25.01 | 25.03 | 27.01 | 26.00 |
| VisualGLM-6B | 20.71 | 18.02 | 28.56 | 23.13 | 22.23 | 23.47 | 22.50 | 22.53 | 23.26 | 22.89 |
| Llava-V1.6-Mistral-7B | 20.39 | 21.62 | 30.33 | 24.99 | 24.01 | 26.70 | 23.39 | 20.96 | 28.62 | 24.73 |
| mPLUG-Owl2.1 | 20.49 | 24.29 | 30.93 | 26.04 | 25.53 | 28.42 | 24.19 | 21.16 | 30.79 | 25.90 |
| InstructBLIP-Vicuna-7B | 21.63 | 18.79 | 27.65 | 23.44 | 21.95 | 23.07 | 23.02 | 19.98 | 26.19 | 23.04 |
| Qwen-VL-Max | 26.71 | 27.37 | 35.84 | 30.65 | 30.22 | 31.14 | 30.13 | 30.27 | 30.82 | 30.54 |
| GPT-4V | 22.48 | 30.92 | 40.84 | 33.00 | 31.47 | 34.32 | 31.42 | 31.10 | 34.13 | 32.59 |
| *+ Heuristic Two-Step mRAG: Retrieving Image First, then Retrieving Web Pages with Question appended to Retrieved Caption* | | | | | | | | | | |
| Qwen-VL-Chat | 19.17 | 20.02 | 28.54 | 23.33 | 22.68 | 23.84 | 22.69 | 24.11 | 22.17 | 23.16 |
| Qwen-7B-Chat | 15.27 | 17.33 | 28.53 | 21.70 | 19.83 | 26.65 | 17.50 | 20.47 | 21.96 | 21.20 |
| Deepseek-VL-7B-Chat | 18.65 | 23.51 | 34.68 | 26.66 | 26.54 | 29.74 | 24.51 | 25.70 | 27.59 | 26.63 |
| VisualGLM-6B | 18.56 | 19.98 | 29.43 | 23.87 | 21.84 | 27.07 | 20.79 | 22.00 | 24.70 | 23.33 |
| Llava-V1.6-Mistral-7B | 18.41 | 20.81 | 34.70 | 26.27 | 23.97 | 30.49 | 22.37 | 19.10 | 32.41 | 25.65 |
| mPLUG-Owl2.1 | 14.10 | 19.28 | 30.24 | 22.77 | 20.74 | 26.96 | 19.02 | 16.24 | 28.41 | 22.23 |
| InstructBLIP-Vicuna-7B | 16.22 | 18.95 | 30.34 | 22.96 | 22.07 | 25.47 | 20.86 | 16.46 | 29.18 | 22.72 |
| Qwen-VL-Max | 24.44 | 30.75 | 43.21 | 34.03 | 33.91 | 38.26 | 31.1 | 32.04 | 36.01 | 33.99 |
| GPT-4V | 20.37 | 33.98 | 48.46 | 36.12 | 36.04 | 40.19 | 33.32 | 32.99 | 39.30 | 36.10 |
| *+ Heuristic Two-Step mRAG: Image Caption First, then Retrieving Web Pages with Question appended to Caption* | | | | | | | | | | |
| Qwen-VL-Chat | 22.05 | 25.87 | 31.84 | 27.58 | 26.21 | 27.46 | 27.06 | 28.81 | 25.57 | 27.21 |
| Qwen-7B-Chat | 14.65 | 21.16 | 28.66 | 22.89 | 21.02 | 23.64 | 21.55 | 16.57 | 28.39 | 22.39 |
| Deepseek-VL-7B-Chat | 21.12 | 27.65 | 36.27 | 29.41 | 29.08 | 29.72 | 29.05 | 26.84 | 31.88 | 29.32 |
| VisualGLM-6B | 19.60 | 21.75 | 33.23 | 25.88 | 25.23 | 27.27 | 24.65 | 22.95 | 28.56 | 25.71 |
| Llava-V1.6-Mistral-7B | 21.09 | 26.41 | 33.87 | 28.20 | 27.25 | 29.76 | 26.71 | 21.66 | 34.42 | 27.94 |
| mPLUG-Owl2.1 | 20.46 | 26.67 | 34.91 | 28.77 | 26.90 | 28.31 | 28.25 | 20.33 | 36.47 | 28.27 |
| InstructBLIP-Vicuna-7B | 24.37 | 28.22 | 35.62 | 30.51 | 29.85 | 30.77 | 30.06 | 22.98 | 35.57 | 30.33 |
| Qwen-VL-Max | 24.27 | 32.93 | 44.03 | 35.04 | 34.97 | 35.16 | 34.92 | 31.1 | 39.05 | 35.02 |
| GPT-4V | 24.90 | 36.74 | 45.76 | 37.23 | 36.94 | 37.82 | 36.70 | 31.65 | 42.81 | 37.15 |
| *Generative Search Engine* | | | | | | | | | | |
| Bing Chat | 27.71 | 32.77 | 32.99 | 31.67 | 30.80 | 35.44 | 28.64 | 29.62 | 32.74 | 31.44 |
| Perplexity AI | 29.62 | 34.69 | 34.88 | 33.67 | 32.81 | 37.46 | 30.67 | 31.59 | 34.80 | 33.51 |
| Gemini | 36.17 | 32.86 | 42.84 | 38.75 | 34.78 | 46.39 | 31.82 | 35.77 | 39.69 | 37.69 |
| *Ours* | | | | | | | | | | |
| OmniSearch (Qwen-VL-Chat) | 35.16 | 40.89 | 45.52 | 41.34 | 40.81 | 42.56 | 40.28 | 39.22 | 43.23 | 41.20 |
| OmniSearch (GPT-4V) | **44.04** | **49.58** | **54.45** | **50.38** | **49.06** | **50.49** | **49.73** | **46.96** | **53.21** | **50.03** |
| *Estimated Upper Bound: + Retrieving Web Pages with Gloden Query* | | | | | | | | | | |
| Qwen-VL-Chat | 37.46 | 46.52 | 52.18 | 46.73 | 45.28 | 47.94 | 45.27 | 43.88 | 48.90 | 46.35 |
| Qwen-7B-Chat | 39.69 | 47.27 | 57.76 | 49.53 | 49.02 | 50.94 | 48.36 | 46.02 | 52.88 | 49.40 |
| Deepseek-VL-7B-Chat | 35.89 | 46.00 | 54.29 | 46.84 | 45.92 | 49.20 | 44.81 | 45.41 | 47.80 | 46.59 |
| VisualGLM-6B | 36.09 | 40.09 | 50.59 | 43.33 | 42.75 | 44.00 | 42.61 | 41.98 | 44.40 | 43.17 |
| Llava-V1.6-Mistral-7B | 39.67 | 48.56 | 56.09 | 49.67 | 47.80 | 52.38 | 47.00 | 43.90 | 54.61 | 49.17 |
| mPLUG-Owl2.1 | 41.19 | 48.66 | 55.83 | 49.63 | 49.17 | 51.77 | 47.97 | 44.49 | 54.68 | 49.51 |
| InstructBLIP-Vicuna-7B | 37.17 | 43.92 | 55.53 | 46.51 | 45.94 | 49.22 | 44.45 | 41.25 | 54.05 | 46.36 |
| Qwen-VL-Max | 42.19 | 53.01 | 56.6 | 51.91 | 50.58 | 52.97 | 50.6 | 49.83 | 53.33 | 51.56 |
| GPT-4V | 45.59 | 54.23 | 60.78 | 55.15 | 52.81 | 54.53 | 54.51 | 51.08 | 58.07 | 54.52 |
| Human Performance | 51.63 | 60.02 | 53.19 | 54.12 | 58.31 | 57.86 | 53.20 | 51.96 | 58.36 | 55.12 |

Table 11: Performances of GPT-4V and OmniSearch(G) with different top-k retrieved content.

| Model | # R.C. | Answer Update Frequency | | | Reasoning Steps | | Visual-Seeking | | Language | | all |
|---|---|---|---|---|---|---|---|---|---|---|---|
| | | fast | slow | never | ≤ 2-hop | > 2-hop | no | yes | zh | en | |
| GPT-4V | None | 17.63 | 27.80 | 40.82 | 30.80 | 28.74 | 31.71 | 29.26 | 26.44 | 34.18 | 30.25 |
| GPT-4V | 1 | 24.07 | 32.6 | 46.11 | 35.47 | 34.98 | 37.98 | 33.54 | 32.91 | 38.02 | 35.33 |
| GPT-4V | 3 | 26.66 | 37.7 | 47.65 | 39.05 | 38.02 | 40.01 | 37.94 | 37.37 | 40.21 | 38.78 |
| GPT-4V | All | 24.90 | 36.74 | 45.76 | 37.23 | 36.94 | 37.82 | 36.70 | 31.65 | 42.81 | 37.15 |
| OmniSearch (G) | 1 | 35.16 | 40.89 | 45.52 | 41.34 | 40.81 | 42.56 | 40.28 | 39.22 | 43.23 | 41.20 |
| OmniSearch (G) | 3 | 39.08 | 49.06 | 52.87 | 48.93 | 45.89 | 52.38 | 44.98 | 48.89 | 48.23 | 48.09 |
| OmniSearch (G) | All | 44.04 | 49.58 | 54.45 | 50.38 | 49.06 | 50.49 | 49.73 | 46.96 | 53.21 | 50.03 |

Table 12: Consistency of Different Metrics Consistency of Different Metrics Consistency of Different Metrics.

| Model | Recall | GPT-based Eval. | Human Eval. | Correlation | | |
|---|---|---|---|---|---|---|
| | | | | #1 & #2 | #1 & #3 | #2 & #3 |
| GPT-4V + mRAG | 40.23 | 37.50 | 42.00 | 0.47 | 0.34 | 0.66 |
| Qwen-VL-Chat + mRAG | 28.46 | 26.50 | 25.00 | 0.49 | 0.41 | 0.69 |
| OmniSearch (G) | 57.56 | 54.00 | 49.00 | 0.45 | 0.38 | 0.63 |
| OmniSearch (Q) | 41.39 | 40.50 | 39.50 | 0.43 | 0.37 | 0.61 |

(2) Unlike GPT-4V, which does not exhibit continual improvement with increased retrieval volume, our OmniSearch demonstrates superior capacity for utilizing extensive retrieved content. This indicates that despite potential noise contained in the retrieved content that may be harmful to question solving, OmniSearch can effectively filters or disregards such disturbances, thereby adequately leveraging the complex and voluminous retrieved content.

(3) For the English question, the performance of both models continues to grow with the increase of retrieved content. This is partly due to the fact that both models use GPT-4V as the backbone, which is inherently more capable in English than in Chinese, and also arises from the fact that Google search is naturally more inclined towards English websites, leading to better support for English questions. This inspires us that in the future, more search tools can be introduce based on language characteristics, such as Bing, Baidu, etc., and even multiple search tools can be utilized to verify or vote for the final answers.

### D.2 CONSISTENCY OF DIFFERENT EVALUATION METRICS

In the main experiment, we employed F1-Recall as an evaluation metric due to its convenience. To demonstrate that it reliably reflects the true capabilities of models, we introduced two supplementary metrics: GPT-based Accuracy and Human-based Accuracy. For these metrics, we presented questions with ground-truth answers to GPT-4V and human evaluators, respectively, asking them to assess the correctness of the model responses and then compute the percentage of correct answers. Table 12 delineates the scores of the different models across these three metrics, as well as their correlation, which is quantified by the Pearson correlation coefficient. The Pearson coefficient ranges from -1 to 1, with 1 signifies a perfect positive correlation and -1 denotes a complete negative correlation. The trends of the different models across these three metrics are entirely consistent, with all coefficients exceeding 0, affirming a positive correlation. This demonstrates that F1-Recall fully reflects model performance. While GPT-based Accuracy and Human-based Accuracy exhibit stronger consistency, which may prove that they are more reliable, F1-Recall remains advantageous as an automated metric, offering significantly lower computational costs and better scalability.

### D.3 SUPPLEMENTARY ANALYSIS OF COMPUTATIONAL COSTS

In Table 13, we further reported the computational cost of OmniSearch on questions with different answer update frequencies (fast, slow, never). The results indicate that:

Table 13: The token cost of OmniSearch on questions with different answer update frequencies.

| Planning Model | Sub-question Solver | Fast | | | Slow | | | Never | | |
|---|---|---|---|---|---|---|---|---|---|---|
| | | Input T. | Output T. | Perf. | Input T. | Output T. | Perf. | Input T. | Output T. | Perf. |
| OmniSearch (G) | OmniSearch (G) | 3098.5 (G) | 466.3 (G) | 44.04 | 3036.5 (G) | 477.0 (G) | 49.58 | 2974.6 (G) | 483.9 (G) | 54.45 |
| OmniSearch (G) | Qwen-VL-Chat | 1268.2 (G) + 2165.4 (Q) | 403.1 (G) + 148.4 (Q) | 38.65 | 1214.1 (G) + 1925.7 (Q) | 358.5 (G) + 112.7 (Q) | 44.68 | 1185.6 (G) + 2138.8 (Q) | 398.2 (G) + 119.3 (Q) | 52.25 |
| OmniSearch (Q) | OmniSearch (Q) | 10258.2 (Q) | 638.1 (Q) | 35.16 | 9874.8 (Q) | 634.2 (Q) | 40.89 | 8866.9 (Q) | 475.3 (Q) | 45.52 |
| OmniSearch (Q) | GPT-4V | 2508.8 (G) + 933.0 (Q) | 269.3 (G) + 547.8 (Q) | 37.14 | 2452.5 (G) + 1000.9 (Q) | 233.6 (G) + 490.5 (Q) | 42.82 | 2209.6 (G) + 1024 (Q) | 329.7 (G) + 606.3 (Q) | 47.48 |

Table 14: The average model latency for question in Dyn-VQA.

| Planning Model | Sub-question Solver | Search Time (s/%) | Inference Time (s/%) | Total Time (s) |
|---|---|---|---|---|
| OmniSearch (G) | OmniSearch (G) | 14.1 (44.8%) | 18.4 (55.2%) | 31.5 |
| OmniSearch (G) | Qwen-VL-Chat | 12.3 (44.6%) | 15.3 (55.4%) | 27.6 |
| OmniSearch (Q) | OmniSearch (Q) | 8.5 (38.3%) | 13.7 (61.7%) | 22.2 |
| OmniSearch (Q) | GPT-4V | 11.8 (45.0%) | 14.4 (55.0%) | 26.2 |

(1) Overall, OmniSearch consumes more tokens for more complex questions, such as those with fast or slow answer updates, primarily because these questions inherently require more retrieval steps.

(2) The difference in resource consumption for questions of varying difficulty is more pronounced for the smaller model OmniSearch (Q). This is due to the behavioral differences between GPT-4V and Qwen-VL. Specifically, as described in the Section 6.1, GPT-4V tends to be more rigorous in question-solving, proactively planning verification retrievals to validate final answers, leading to retrieval processes exceeding three steps even for some relatively easier questions.

(3) By comparing rows 1 and 2 in the Table 13, we can observe that after replacing OmniSearch (G)'s Sub-question Solver with a smaller Qwen-VL-Chat, the total token consumption of Qwen-VL-Chat and GPT-4V is stil comparable to the original model. This suggests that OmniSearch (G) effectively offloads computational burden to smaller models without a significant drop in overall performance.

Additionally, we also reported the average latency for each question in Table 14. The results show that substituting some modules in OmniSearch with smaller models effectively reduces latency. Moreover, the ratio of search time to model inference time is roughly 2:3, indicating significant potential for optimization in both aspects. It is important to note that latency is a complex system engineering issue that involves not only the model's complexity but also factors network configuration of search APIs, caching strategies for retrieval content, inference model acceleration, and hardware FLOPS, etc.

# E   ANALYSIS EXPERIMENTS ON DYN-VQA DATASET

## E.1   MODEL PERFORMANCE ON OTHER VQA DATASETS

Table 15 presents the model performances on various VQA datasets, highlighting several observations:

(1) The original GPT-4V achieves an average performance exceeding 74 on previous datasets, approaching human-level proficiency in the Table 3. Conversely, its performance poorly on our Dyn-VQA with a significantly lower F1-reacll of 30.25, substantially lagging behind human capabilities. This discrepancy primarily arises because MLLMs have internalized much of the knowledge pertinent to traditional VQA datasets, where many questions rely on common-sense knowledge. For instance, VQAv2 frequently poses questions about object properties or intentions behind actions, which are relatively specialized by GPT-4V.

(2) The performance of the heuristic mRAG method is unstable on different types of questions (datasets). It impairs the effectiveness of GPT-4V on all previous datasets, especially on the InfoSeek where the decline is over 10 points. Through case analysis, we found that the heuristic mRAG struggles predominantly with questions of images depicting buildings, specific flora and fauna, which the image caption model fails to describe accurately. Consequently, the search engine yields "shallow"

Table 15: Model performance comparison on different VQA datasets. Heuristic mRAG in Table represents image caption-based mARG.

| Model | VQAv2 | A-OKVQA | InfoSeek | Dyn-VQA |
|---|---|---|---|---|
| *Original MLLMs* | | | | |
| GPT-4V | 68.00 | 83.63 | 70.44 | 30.25 |
| *+ Heuristic Two-Step mRAG* | | | | |
| GPT-4V | 65.36 | 81.00 | 58.64 | 37.15 |
| *Ours* | | | | |
| OmniSearch (G) | 70.34 | 84.12 | 71.48 | 50.03 |

knowledge, i.e., content that is relevant to the question topic but is actually irrelevant to question, and instead misleads the original model. Moreover, given that InfoSeek is automatically generated from Wikipedia and characterized by homogenous question types, a substantial proportion of these problematic questions magnifies the disability of heuristic mRAG. On the contrary, the questions of VQAv2 and A-OKVQA typically inquire about common-sense knowledge, which is quite different from the real-world knowledge on the Internet, therefore the retrieved content instead has a minor (but still present) negative impact on the model.

(3) Our OmniSearch method achieved steady gains on each dataset. Even for datasets such as VQAv2 and A-OKVQA, which demand less extensive real-world knowledge from the Internet, OmniSearch still achieved slight growth. Since OmniSearch potentially makes search determinations, allowing it to avoid unnecessary retrieval interference with the intrinsic model understanding of questions that actually do not benefit from external knowledge augmentation. OmniSearch proves more adaptable and robust to diverse question types compared to the heuristic mRAG.

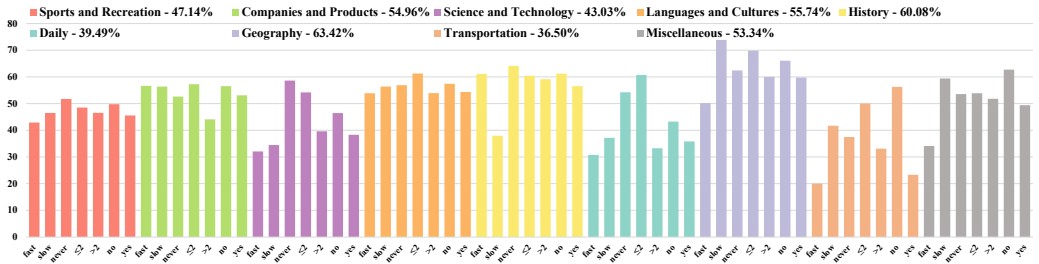

Figure 7: The performance of OmniSearch on different domains. Each category is represented by 7 bars of the same color, representing questions of different categories (in order fast, slow, never, ≤2-hop, >2-hop, no, yes).

## E.2 OMNISEARCH PERFORMANCE ON DIFFERENT DOMAINS

Figure 7 displays the performance of OmniSearch across various domains, as an complement to Figure 4 in the main content. We find that in most domains, the heights of the bars follow a similar pattern: 1st bar (fast) < 2nd bar (slow) < 3rd bar (never), 4th bar (less than 2-hop) > 5th bar (more than 2-hop), 6th bar (not require external visual knowledge) > 7th bar (require external visual knowledge). This indicates that OmniSearch generally underperforms on questions necessitating more complex retrieval processes. There are exceptions in data-sparse domains such as Transportation, where OmniSearch also shows relatively poor performs in these domains.

## E.3 CASE STUDY

Figure 8 presents several intriguing case studies. In the first case on the left, the model dynamically adjusted its retrieval strategy multiple times after an initial retrieval failure, demonstrating OmniSearch's ability to adapt to complex retrieval environments and actively adjust its strategy based on environmental feedback. However, the subsequent two cases highlight different challenges faced by OmniSearch. In the middle example, after the model estimated the current GDP of South

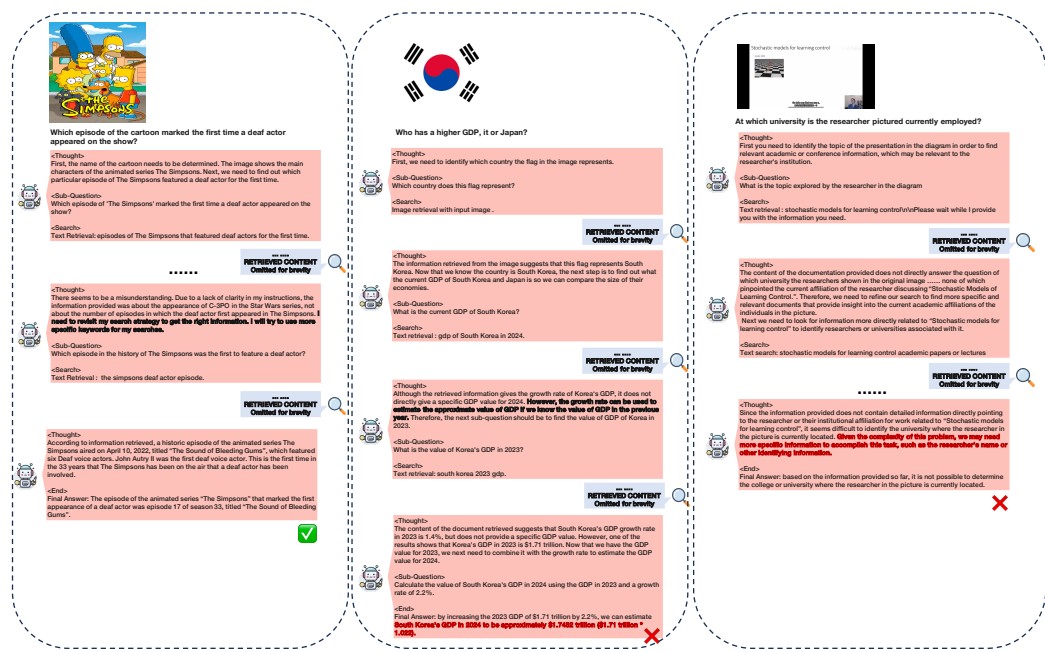

Figure 8: Case study with successful case and two bad cases.

Korea by changing the direct search strategy to an indirect one with the growth rate and the previous year's GDP, it should have proceeded to retrieve Japan's current GDP. Instead, OmniSearch abruptly ceased its thought and retrieval. This indicates that OmniSearch struggles with questions requiring extended reasoning chains and tends to lose track of the original question and preceding information. In the right case, the initial planned retrieval strategy is incorrect from the start, as the main body of the image is a PowerPoint slice, and the person involved in the question only occupies a small space in the bottom right corner of the image. The OmniSearch focuses on the wrong visual evidence and gets caught in a "thinking trap". Ideally, the model should first search the image to find out that it is a screenshot of a course video, then view the video to find out the name of the speaker, and further search the web page to find out information about his academic institution. Another alternative is to get the exact search region, i.e., the bottom-right corner of the image, through image object recognition and image cropping. Then the person information is obtained based on the caption of the retrieved image. However, both approaches cannot be perfectly achieved by the current OmniSearch, which does not support such complex and fine-grained retrieval. These failed cases bring us significant insights and inspirations: firstly, how to solve the question requiring long context knowledge is worth studying, we analysed 100 error cases, and found that 73 of them encounter the issue of partially containing the correct answer, but the OmniSearch can't complete the full retrieval process due to the long context. On the one hand, we need to improve the maximum length of the context window of MLLMs, on the other hand, how to denoise, compress, and summarize the context is the direction that the sub-problem solver of OmniSearch can be improved. Secondly, advancing more precise retrieval techniques and incorporating a broader range of retrieval tools is an urgent research to be carried out.

