# OpenReview forum: "Benchmarking Multimodal Retrieval Augmented Generation with Dynamic VQA Dataset and Self-adaptive Planning Agent"
_ICLR.cc/2025/Conference — ICLR 2025 Poster_

### Official Review · Reviewer_t3io · 2024-11-02

**Soundness:** 2
**Presentation:** 3
**Contribution:** 2
**Rating:** 5
**Confidence:** 4

**Summary:**

In this paper, the authors propose OmniSearch to address the limitations of existing Multimodal Retrieval Augmented Generation (mRAG) methods: (1) non-adaptive retrieval queries and (2) overloaded retrieval queries. Additionally, considering these limitations cannot be adequately reflected in current knowledge-seeking visual question answering datasets, the authors construct the Dyn-VQA dataset. It consists of three types of “dynamic” questions and requires complex knowledge retrieval strategies. Experiments demonstrate the effectiveness of the proposed OmniSearch.

**Strengths:**

1.	The proposed Dyn-VQA dataset bridges the gap in the existing VQA-based mRAG benchmarks. It is oriented towards real-world problems and emphasizes the ability to retrieve dynamic knowledge during the question-answering process.
2.	This paper provides the analysis of the statistical information, data quality and diversity of the proposed Dyn-VQA dataset.
3.	The experiment results show the effectiveness of the proposed OmniSearch method.

**Weaknesses:**

1.	The scale of the proposed Dyn-VQA dataset is small, containing only 1,500 questions. It is significantly fewer than existing knowledge-seeking VQA datasets.
2.	Although the proposed OmniSearch shows the potential in application, the innovation is limited.
3.	As mentioned in the case study, the OmniSearch method struggles with questions that require long-term reasoning chains and fine-grained questions.

**Questions:**

1.	The authors may consider expanding the scale of the Dyn-VQA dataset.
2.	The Dyn-VQA dataset considers two languages (Chinese and English). I wonder if there are impacts on different LLM or MLLM models? For example, some models only support Chinese and English, but others support more languages.
3.	The authors should explain how OmniSearch dynamically decomposes the sub-questions and how it determines whether a question has been resolved?
4.	The authors should explain the difference between OmniSearch and the chain-of-thought method.
5.	In experiments, the authors may consider evaluating the effectiveness of OmniSearch based on more LLMs or MLLMs.

---

> ### Author Response · Authors · 2024-11-21
> **Response to Reviewer t3io**
>
> We sincerely appreciate your valuable feedback. It seems that there are some misunderstanding in the review. Here are some clarifications:
>
> - ***the scale of the proposed Dyn-VQA dataset is small***
>
> **Answer:**
>
> Regarding the dataset size, we discuss this in Section 3.2 (lines 215-228). While our dataset cannot be directly compared to others in terms of scale, it significantly exceeds others in terms of quality and the human effort involved in its creation:
>
> a. **Data Quality:** our dataset is entirely manually curated, whereas many other datasets are generated using templates or rule-based methods, which sacrifice the dataset quality. Additionally, our dataset covers a rich variety of domains and categories to ensure unbiased representation.
>
> b. **Dynamic Updates:** unlike static datasets, ours requires continuous, month-by-month human effort for updates and maintenance to ensure quality. We believe that few organizations can easily commit such labor and passionate. To date, we have annotated three versions of the dataset.
>
> Besides, similar to datasets like ViQuAE and S3VQA, whose data volume is also on the order of thousands, we believe that a dataset of this scale is sufficient to reflect the performance of RAG models.
>
> --------
>
> - ***Innovation is limited***
>
> **Answer:**
>
> In this paper, we reveal that previous multimodal RAG methods primarily relied on heuristic retrieval approaches that fail to provide correct, relevant knowledge for dynamic questions effectively. To our knowledge, **OmniSearch is the first multimodal retrieval-enhanced agent** that dynamically plans a rational retrieval path for input multimodal questions and provides accurate knowledge.
>
> Additionally, you mentioned concerns about the differences between OmniSearch and Chain of Thought (CoT) methods. The **fundamental distinction** between OmniSearch, as a multimodal agent, and CoT is its ability to utilize tools, interact with the environment, and response to the environment [1]. In contrast, CoT methods primarily stimulate the model's inherent logical reasoning capabilities through prompts. The CoT approach is unable to decouple intermediate processes, therefore can not be integrated with retrieval tools. We have added this discussion in the revised paper.
>
> **Besides the innovation in methodology,** we constructed Dyn-VQA, which requires more complex retrieval processes compared to existing benchmarks, allowing for a more realistic evaluation of mRAG performance. Therefore, we believe the novelty criticism is somewhat unfounded, **as other reviewers also point out the novelty in their reviews.**
>
> Reference:
>
> [1] Zhang, et al. Igniting Language Intelligence: The Hitchhiker's Guide From Chain-of-Thought Reasoning to Language Agents. arXiv 2024.
>
> -----------
>
> - ***OmniSearch struggles with some questions in case study***
>
> **Answer:**
>
> We acknowledge that OmniSearch does not perform exceptionally in some cases, just like all AI models are not perfect. However, overall, OmniSearch significantly surpasses the performance of GPT4-V combined with various heuristic retrieval methods and various commercial search engine. The purpose of the case study is more to inspire future work.
>
> --------
>
> - ***The impacts of different language***
>
> **Answer:**
>
> we fully agree with your suggestion regarding multilingual aspects. Multilingual support is indeed a valuable future direction, particularly for exploring cross-lingual generalization, language gaps, and the benefits of mainstream languages for low-resource languages in the era of LLMs. We plan to explore this broad topic in future work. It is worth noting that most VQA datasets are monolingual, while ours is bilingual, providing a richer evaluation perspective. The dataset from different language perspectives is also analyzed in Appendix D1 (lines 1112-1118).
>
> -------
>
> - ***Explain how OmniSearch decomposes the question and decides on the end.***
>
> **Answer:**
>
> The prompts used by OmniSearch can be found in Appendix C.4, which clearly shows how OmniSearch decomposes questions. When OmniSearch outputs \<Final Answer\>, it indicates that the question has been resolved.
>
> --------
>
> - ***Consider evaluating the effectiveness of OmniSearch based on more LLMs or MLLMs.***
>
> **Answer:**
>
> We appreciate your suggestion to evaluate OmniSearch with more LLMs and MLLMs. GPT4-V and Qwen-VL were chosen as the backbone for OmniSearch. They represent the leading models among open-source small models and proprietary large models, respectively, and we believe they are sufficiently representative. Additionally, Table 11 demonstrates the results for more than 10 models.

---

> ### Author Response · Authors · 2024-11-24
> **Inquiry Regarding Review Feedback for Paper Revisions**
>
> Dear Reviewer,
>
> Understanding your busy schedule, we wanted to check if the revisions and details provided have addressed your concerns. Please let us know if any issues remain or further clarification is needed. Your feedback is invaluable to us for improving our work.
>
> Thank you for your attention.

---

> ### Author Response · Authors · 2024-11-27
> **Friendly reminder to reviewer t3io**
>
> Based on your feedback, we have **clarified the novelty of our method and further discussed the dataset size**. We hope these analyses address your concerns.
>
> Considering that the rebuttal discussion phase for ICLR is about to end, we sincerely hope to receive your further feedback so that we can continue our discussion. Once again, thank you for your hard work and selfless help.

---

### Official Review · Reviewer_ia8N · 2024-11-03

**Soundness:** 2
**Presentation:** 3
**Contribution:** 2
**Rating:** 6
**Confidence:** 4

**Summary:**

The authors focuses on a dynamic visual question answering (VQA) task, where the answers can change over time (e.g., "What is his latest film?"). The authors introduce a new dynamic VQA benchmark to investigate the limitation of current multimodal retrieval augmented generation (mRAG) methods. The authors propose a new retrieval method, OmniSearch, which is based on the self-adaptive planning agent.

The contributions of this paper include (i) a new benchmark for the dynamic VQA task and (ii) a new approach---a self-adaptive planning agent---to improve task peformance.

**Strengths:**

- **Introduction of a new VQA benchmark**: The authors focus on "dynamic" visual questions, where answers can change over time. This type of question is frequently encountered in real-world scenarios but is underrepresented in existing VQA datasets. The authors propose a new benchmark featuring dynamic visual questions that reflect the complexity of real-world inquiries.

- **Proposal of a new mRAG approach**: The author introduces a self-adaptive retrieval agent that plan seach retrieval actions in real time and demonstrate its effectivness on dymanic VQA tasks. This approach is designed as a plug-and-play module that can be incorporated into various MLLMs, showing its applicability.

- **Presentation**: The paper is clearly written and easy to follow.

**Weaknesses:**

- **Sustainability of Benchmark Accuracy**: Since the answers to certain questions in this benchmark may change over time, there is a risk that the answers will become **outdated** after the benchmark is publicly released. This raises concerns about how to ensure accurate model evaluation in such cases (i.e., when the benchmark's ground-truth answers no longer reflect current information). How will the benchmark address this issue to continue providing reliable, up-to-date evaluations?

- **Unclear Definition of "Hops" in Questions**: The benchmark offers "multi-hop" visual questions, which the authors claim are often missing in existing VQA benchmarks. However, the paper lacks clarity in defining a "hop". While the authors consider one reasoning step as one hop, they do not specify how they define a single reasoning step. Interpretations of reasoning steps can vary; for example, some may consider a particular reasoning process as requiring two steps, while others might view it as involving only one step. Setting clear criteria for defining a reasoning step is essential.

- **Missing Related Work**: While the paper focuses on multi-hop visual questions, it does not address recent work in this area. For example, it omits the citation, "Kil et al., II-MMR: Identifying and Improving Multi-modal Multi-hop Reasoning in Visual Question Answering, ACL'24."

**Questions:**

Please see the weakness of the paper.

---

> ### Author Response · Authors · 2024-11-21
> **Response to Reviewer ia8N**
>
> We sincerely thank you for your review and feedback. We will make every effort to address your concerns and kindly request you to potentially increase the score based on the following points:
>
> -------
>
> - *Sustainability of Benchmark Accuracy*
>
> **Answer:**
>
> We discuss the regular updates of our dataset in Appendix A.4. As you noted, our dataset is not static and requires continuous human effort for updates and maintenance to ensure quality. While few organizations can easily achieve this, we commit to maintaining regular updates. Currently, we have annotated three versions of the dataset.
>
> --------
>
> - *Definition of "Hops" in Questions*
>
> **Answer:**
>
> The definition of "hops" in our work aligns with past QA research, referring to the number of knowledge rationales required to answer a question. For example, the question "What is the altitude of the mountain in the image?" typically requires two knowledge rationales: the name of the mountain and the corresponding altitude.
>
> While different annotators may have inconsistencies for some complex questions, these issues usually involve more than two knowledge rationales. For example, arguing about whether the quesiton is a three or four hop question. However, **this does not affect the annotation accuracy or our motivation:** most VQA dataset have fixed two-hop questions. We also **hired two extra quality control personnel** to ensure consistent and unbiased annotation. Moreover, as shown in Table 3, the average search count of each DynVQA question involved in human question answering exceeds three, significantly surpassing other datasets, which also indicates the challenge of Dyn-VQA in terms of multi-hop question.
>
> ----------
>
> - *Missing Related Work*
>
> Our work primarily focuses on multimodal retrieval-augmented generation (mRAG). Mainstream mRAG methods are generalized into four categories in method section and are also introduced thoroughly in related work section. To the best of our knowledge, OmniSearch is the first multimodal RAG agent.
>
> The work you mentioned primarily focuses on multi-hop QA for commonsense knowledge (e.g., "Is the temperature high or low in the region (ref. to a refrigerator) where the bottle is located?"), which does not use the mRAG method. But the dataset it used, like A-OKVQA, is relevant to our dataset, which is also discussed in our paper. Hence, we added the citation to Kil et al. (ACL'24) and some multi-hop QA methods in the related work section. Thanks for your suggestions.

---

> ### Author Response · Authors · 2024-11-24
> **Inquiry Regarding Review Feedback for Paper Revisions**
>
> Dear Reviewer,
>
> Understanding your busy schedule, we wanted to check if the revisions and details provided have addressed your concerns. Please let us know if any issues remain or further clarification is needed. Your feedback is invaluable to us for improving our work.
>
> Thank you for your attention.

---

> > ### Comment · Reviewer_ia8N · 2024-11-26
> >
> > Thank you for your response. The authors addressed my concerns: I’ve raised the score, hoping they’ll update the benchmark regularly.

---

> > > ### Author Response · Authors · 2024-11-27
> > >
> > > Thank you for the recognition and dedication. We are committed to continuing to maintain the dataset with the same effort.

---

### Official Review · Reviewer_Z1ru · 2024-11-03

**Soundness:** 4
**Presentation:** 3
**Contribution:** 3
**Rating:** 8
**Confidence:** 4

**Summary:**

This paper introduces Dyn-VQA a new dataset for benchmarking multimodal retrieval-augmented generation (mRAG) methods. The dataset is constructed specifically with tasks that require adaptive retrieval methods that can handle changing answers, multi-hop reasoning and multi-modal knowledge. The authors run several baselines/benchmarks on this dataset with different MLLM models, and mRAG methods. The authors also introduce OmniSearch, a self-adaptive planning agent designed to overcome the limitations of current mRAG methods (the limitations this dataset addresses), and show a performance gain from OmniSearch.

**Strengths:**

1. They introduce a strong, novel dataset, Dyn-VQA, which offers a new, and uniquely hard multi-modal retrieval challenge for adaptable, multi-hop retrievals - which mimics real world settings well.

2. Strong experimental section - they benchmark this dataset with several MLLMs, and several types of mRAG methods.

3. They introduce the OmniSearch method which performs very well on this task.

4. The paper is mostly clearly written and well motivated.

**Weaknesses:**

1. The dataset has a strong motivation, however, the abstract and introduction could more clearly address the concepts of (as you labeled them) (1) Non-adaptive Retrieval Queries and (2) Overloaded Retrieval Queries. Clarifying these issues up front would help position and motivate the work better - and I felt they weren't so clearly explained.

2. There could be more discussion around the scalability and computational cost of OmniSearch to provide a better sense of its applicability in real-world/time settings.

**Questions:**

1. Could you say more about the computational costs of OmniSearch specifically as it scales with multi-step, dynamic retrievals?

2. It is interesting that no questions was correctly answered by all models - and in general Figure 5 is really interesting to see - do you have any intuition for this distribution? Could it be used to sort of profile what tasks a specific LLM is good at?

---

> ### Author Response · Authors · 2024-11-21
> **Response to Reviewer Z1ru - Part I**
>
> Thank you for your feedback and valuable suggestions. We will make every effort to address your concerns:
>
> ---------
>
> - ***Explanation of Non-adaptive Retrieval Queries and Overloaded Retrieval Queries***
>
> **Answer:**
>
> We appreciate your feedback on the need for clearer explanations. We have provided further clarification and incorporated these into the revised paper:
>
> - **Non-adaptive Retrieval Queries** refer to the fixed retrieval processes and query structures of heuristic mRAG methods. These inflexible retrieval strategies fail to adapt to evolving contexts or intermediate findings within a question, hindering the model from re-retrieving to further comprehend, verify, or rethink the question. For example, in Figure 2, question (a) asks, "What is his (the actor Cillian Murphy's) latest film?" A fixed retrieval process returns multiple relevant films, but heuristic methods fail to construct further retrieval based on the retrieved content to distinguish between the sequence of different films.
> - **Overloaded Retrieval Queries** refer to heuristic retrieval methods refer to heuristic mRAG methods merely format a single query by concatenating textual descriptions of objects in images with input questions. A single query carries multiple retrieval aspects, leading to ambiguous retrieval and influx of superficially relevant knowledge yet not essential to the question solving. For example, in Figure 2, question (c) asks, "Which one of them (two actors, Ling Jia, and Teng Shen) grossed more?" Heuristic methods might generate a single query like "Ling Jia, Teng Shen, Which one of them grossed more?", which contains the intent to retrieve box office information for both actors. This mixed query conversely fails to provide precise knowledge for each individual aspect.
>
> ---------
> - ***Phenomenon in Figure 5***
>
> **Answer:**
>
> The phenomenon depicted in Figure 5 is indeed intriguing, showing that different MLLMs perceive the difficulty of the same question differently. We believe this difference stems from several factors:
>
> 1. **Capability of Image Encoder:** Different MLLMs use different pre-trained image encoders to extract visual features. Due to variations in pre-training tasks, these encoders differ in their ability to understand image semantics. For example, BEIT excels in leveraging finer-grained image semantics, such as object and background information, by pre-training on masked image modeling. In contrast to the VIT model which learns the overall image semantics through image classifying images in the Image Net dataset. And the image encoder of BLIP also captures richer semantics than the VIT model owing to its linkage with the text modality.
> 2. **Instruction Training Data:** The instruction data used to train different MLLMs vary significantly, which affects the model's inner capabilities. For example, Qwen-VL has been trained with a relatively greater volume of OCR Instruction data compared to other MLLMs, making it excel at tasks involving text within images.
> 3. **Dataset Characteristics:** Dyn-VQA encompasses a wider variety of question types compared to existing VQA datasets, requiring models to possess more comprehensive capabilities. An ideal model must have robust abilities in understanding input questions and analyzing multimodal retrieval content, such as multi-image understanding, commonsense reasoning, and cross-modal inference.

---

> ### Author Response · Authors · 2024-11-21
> **Response to Reviewer Z1ru - Part II**
>
> --------
>
> - *Computational Costs of OmniSearch*
>
>   **Answer:**
>
>   We further reported the computational cost of OmniSearch on questions with different answer update frequencies (fast, slow, never). The results indicate that:
>
>   1. Overall, OmniSearch consumes more tokens for more complex questions, such as those with fast or slow answer updates, primarily because these questions inherently require more retrieval steps.
>   2. The difference in resource consumption for questions of varying difficulty is more pronounced for the smaller model OmniSearch (Q). This is due to the behavioral differences between GPT-4V and Qwen-VL. Specifically, as described in the Section 6.1, GPT-4V tends to be more rigorous in question-solving, proactively planning verification retrievals to validate final answers, leading to retrieval processes exceeding three steps even for some relatively easier questions.
>   3. By comparing rows 1 and 2 in the Table, we can observe that after replacing OmniSearch (G)'s Sub-question Solver with a smaller Qwen-VL-Chat, the total token consumption of Qwen-VL-Chat and GPT-4V is still comparable to the original model. This suggests that OmniSearch (G) effectively offloads computational burden to smaller models without a significant drop in overall performance.
>
>   | Planning Model | Sub-question Solver | Fast Input Token        | Fast Output Token     | Fast Performance | Slow Input Token        | Slow Output Token     | Slow Performance | Never Input Token       | Never Output Token    | Never Performance |
>   | -------------- | ------------------- | ----------------------- | --------------------- | ---------------- | ----------------------- | --------------------- | ---------------- | ----------------------- | --------------------- | ----------------- |
>   | OmniSearch (G) | OmniSearch (G)      | 3098.5                  | 466.3                 | 44.04            | 3036.5                  | 477.0                 | 49.58            | 2974.6                  | 483.9                 | 54.45             |
>   | OmniSearch (G) | Qwen-VL-Chat        | 1268.2 (G) + 2165.4 (Q) | 403.1 (G) + 148.4 (Q) | 38.65            | 1214.1 (G) + 1925.7 (Q) | 358.5 (G) + 112.7 (Q) | 44.68            | 1185.6 (G) + 2138.8 (Q) | 398.2 (G) + 119.3 (Q) | 52.25             |
>   | OmniSearch (Q) | OmniSearch (Q)      | 10258.2                 | 638.1                 | 35.16            | 9874.8                  | 634.2                 | 40.89            | 8866.9                  | 475.3                 | 45.52             |
>   | OmniSearch (Q) | GPT-4V              | 2508.8 (G) + 933.0 (Q)  | 269.3 (G) + 547.8 (Q) | 37.14            | 2452.5 (G) + 1000.9 (Q) | 233.6 (G) + 490.5 (Q) | 42.82            | 2209.6 (G) + 1024 (Q)   | 329.7 (G) + 606.3 (Q) | 47.48             |
>
>   Additionally, we also reported the average latency for each question. The results show that substituting some modules in OmniSearch with smaller models effectively reduces latency. Moreover, the ratio of search time to model inference time is roughly 2:3, indicating significant potential for optimization in both aspects. It is important to note that latency is a complex system engineering issue that involves not only the model's complexity but also factors network configuration of search APIs, caching strategies for retrieval content, inference model acceleration, and hardware FLOPS, etc.
>
>   | Planning Model | Sub-question Solver | Search Time (s/%) | Inference Time (s/%) | Total Time (s) |
>   | -------------- | ------------------- | ----------------- | -------------------- | -------------- |
>   | OmniSearch (G) | OmniSearch (G)      | 14.1 (44.8%)      | 18.4 (55.2%)         | 31.5           |
>   | OmniSearch (G) | Qwen-VL-Chat        | 12.3 (44.6%)      | 15.3 (55.4%)         | 27.6           |
>   | OmniSearch (Q) | OmniSearch (Q)      | 8.5 (38.3%)       | 13.7 (61.7%)         | 22.2           |
>   | OmniSearch (Q) | GPT-4V              | 11.8 (45.0%)      | 14.4 (55.0%)         | 26.2           |
>
> ------

---

> > ### Comment · Reviewer_Z1ru · 2024-11-24
> > **Reviewer Response**
> >
> > Thank you for all the clarification and additional experiments/information.
> >
> > I feel you have addressed all of my concerns.
> >
> > I also see the points raised by other reviewers regarding the scale of the dataset and the necessity of updating it regularly. However, I feel the authors did a good job of clarifying these points, and I agree with them that having a small-scale but 100% human curated dataset is a real contribution - given the importance of having high-quality data, especially for benchmarks.
> >
> > I am leaving my score the same, as I think this is good work and should be accepted.

---

> > > ### Author Response · Authors · 2024-11-25
> > >
> > > Dear Reviewer:
> > >
> > > We sincerely thank you for your positive feedback on our paper, which has greatly motivated us.
> > >
> > > When convenient, could you kindly consider revising your confidence score based on the clarifications provided? Your insights are invaluable to us. Thanks again.

---

> > > > ### Comment · Reviewer_Z1ru · 2024-11-25
> > > >
> > > > After all the feedback and additional studies I do feel more confident in my initial score, so yes I have revised my confidence score accordingly.
> > > >
> > > > Thank you again for all the clarification.

---

### Official Review · Reviewer_6Ev1 · 2024-11-03

**Soundness:** 3
**Presentation:** 3
**Contribution:** 2
**Rating:** 6
**Confidence:** 3

**Summary:**

The paper introduces a new dataset: Dyn-VQA, that fills the gap in existing multi-modal RAG benchmarks by including questions that require varied and dynamic retrieval strategies, 1.5k samples for 9 domains, 2 languages and 3 categories of questions, including rapidly changing knowledge, multimodal knowledge required problem and multi-hop reasoning in mVQA, which is a very essential practical problem in VLM driven search engines.
The paper also presents: OmniSearch, a self-adaptive planning agent decomposes complex questions into sub-questions and dynamically plans retrieval actions to address these challenges. The authors conducted extensive experimental results show that OmniSearch improves performance over traditional heuristic mRAGs.

**Strengths:**

- Originality: The authors have an innovative and practical focus on the gap in existing benchmarks, especially the dynamic retrieval, and multi-modal multi-hop questions. This dataset is extensively curated manually and reflects some real-world complexities.

- Clarity: The paper provides clear definitions, comprehensive descriptions of the dataset construction, and detailed experimental setups. Included some comparison with baselines and highlights the uniqueness and impact of OmniSearch.

- Quality: The authors conducted extensive ablation experiments to compare the performance of OmniSearch.

- Significance: This work provides valuable insights into dynamic planning agent for RAG in multimodal LLMs, focus on a crucial challenge for advancing AI's real-world applicability in mVQA tasks.

**Weaknesses:**

- The dataset's limited size of 1.5k samples, with only 178 questions covering all three challenging categories, raises questions about whether its complexity and diversity are sufficient to benefit the broader research community.

- In data curation part, the dataset only includes English and Chinese, and the authors filtered intractable instances that doesn't translate well, this might limit the dataset's diversity or introduce bias. if more languages are included, and examples are elaborated on how human are correcting Google Translate API, the dataset would be more convincing to be used to evaluate model's generalizability and performance on culturally specific questions that might present real-world challenges.

- For the omniSearch agentic flow, there is no discussion of the latency or computational overhead associated with the multi-step retrieval process.

- The authors seem to not ablate the feedback generation effectiveness in the experiment sections.

- In the experiment part, no reasoning-based commercial search engine such as GPT-4o are included as baselines.

- Figure 3 seems to have a typo in the question, what's the price of this car rather than what's the price on this car?

**Questions:**

- Can the authors elaborate more on the process of how the AI researchers are selected and trained to curate dataset, and how the distribution is defined?
- Can the authors conduct experiments with GPT-4o and also include the latency evaluations? Would also love to see how the feedback generation are effective in other baselines.

---

> ### Author Response · Authors · 2024-11-21
> **Response to Reviewer 6Ev1 - Part I**
>
> We sincerely appreciate your affirmation of the paper soundness, and kindly ask you to consider increasing the score based on the following clarification:
>
> ----------
>
> - *The dataset's limited size of 1.5k samples*
>
> **Answer:**
>
> Regarding the dataset size, we discuss this in Section 3.2 (lines 215-228). While our dataset cannot be directly compared to others in terms of scale, it significantly exceeds others in terms of quality and the human effort involved in its creation:
>
> a. **Data Quality:** our dataset is entirely manually curated, whereas many other datasets are generated using templates or rule-based methods, which sacrifice the dataset quality. Additionally, our dataset covers a rich variety of domains and categories to ensure unbiased representation.
>
> b. **Dynamic Updates:** unlike static datasets, ours requires continuous, month-by-month human effort for updates and maintenance to ensure quality. We believe that few organizations can easily commit such labor and passionate. To date, we have annotated three versions of the dataset.
>
> Besides, similar to datasets like ViQuAE and S3VQA, whose data volume is also on the order of thousands, we believe that a dataset of this scale is sufficient to reflect the performance of RAG models.
>
> ------
>
> - *the authors filtered intractable instances that doesn't translate well, this might limit the dataset's diversity or introduce bias*
>
> **Answer:**
>
> We believe that "filtering instances that doesn't translate well" does not introduces bias. On the contrary, this filtering strategy enhances the evaluation accuracy. For example, consider a question like, "What is the latest work of the author of this book in the picture?" (referring to a local Chinese author). This question is appropriate in the Chinese context but not in the English context, since his books have neither English versions nor English introductions. Hence, we just simply discard the English question in this case, which naturally would not be asked in English by the users.
>
> Meanwhile, we fully agree with your suggestion regarding multilingual aspects. Multilingual support is indeed a valuable future direction, particularly for exploring cross-lingual generalization, language gaps, and the benefits of mainstream languages for low-resource languages in the era of LLMs [1,2]. We plan to explore this broad topic in future work. It is worth noting that most VQA datasets are monolingual, while ours is bilingual, providing a richer evaluation perspective. The dataset from different language perspectives is also analyzed in Appendix D1 (lines 1112-1118).
>
> Reference:
>
> [1] Geigle, et al. Babel-ImageNet: Massively Multilingual Evaluation of Vision-and-Language Representations. ACL 2024.
>
> [2] Qin, et al. Multilingual Large Language Model: A Survey of Resources, Taxonomy and Frontiers. arXiv 2024.
>
> -------
>
> - *latency or computational overhead of OmniSearch*
>
> **Answer:**
>
> In Table 6, we have compared the token consumption (i.e., computational overhead) and corresponding performance of different models. We observe that the full OmniSearch model consumes more tokens because solving problems correctly inherently requires more retrieval and LLM thinking processes.
>
> Here, we also reported the average latency for each question. The results show that substituting some modules in OmniSearch with smaller models effectively reduces latency. Moreover, the ratio of search time to model inference time is roughly 2:3, indicating significant potential for optimization in both aspects. It is important to note that latency is a complex system engineering issue that involves not only the model's complexity but also factors network configuration of search APIs, caching strategies for retrieval content, inference model acceleration, and hardware FLOPS, etc.
>
> | Planning Model | Sub-question Solver | Search Time (s/%) | Inference Time (s/%) | Total Time (s) |
> | -------------- | ------------------- | ----------------- | -------------------- | -------------- |
> | OmniSearch (G) | OmniSearch (G)      | 14.1 (44.8%)      | 18.4 (55.2%)         | 31.5           |
> | OmniSearch (G) | Qwen-VL-Chat        | 12.3 (44.6%)      | 15.3 (55.4%)         | 27.6           |
> | OmniSearch (Q) | OmniSearch (Q)      | 8.5 (38.3%)       | 13.7 (61.7%)         | 22.2           |
> | OmniSearch (Q) | GPT-4V              | 11.8 (45.0%)      | 14.4 (55.0%)         | 26.2           |
>
> Additionally, we supplemented the detailed computational cost of OmniSearch on questions with different answer update frequencies (fast, slow, never), which can be found in the response to Reviewer Z1ru.
>
> ---------------

---

> ### Author Response · Authors · 2024-11-21
> **Response to Reviewer 6Ev1 - Part II**
>
> - *Ablation on the feedback generation effectiveness*
>
> **Answer:**
>
> We have indeed conducted ablation studies on the effectiveness of feedback generation. The feedback in OmniSearch is generated by the sub-question solving model. In Table 6, we compare the performance when using different MLLMs (GPT-4V, Qwen-Vl) as sub-question solvers, i.e., feedback generation effectiveness. We find that overall performance is positively correlated with the sub-question solver's performance. However, the quality of the Planning Model has a more significant impact on overall performance.
>
> -------
>
> - *No GPT-4o as reasoning-based commercial search engine*
>
> **Answer:**
>
> The primary improvement of GPT-4o is in inference speed compared to GPT-4, and its performance is roughly on par with the GPT-4V model. We report its results below. We can observe that the original performance of GPT-4o surpasses that of GPT-4V; however, when equipped with OmniSearch, the performance of GPT-4V is significantly enhanced. Besides, the OmniSearch framework is compatible with arbitrary MLLMs, and we believe that the performance of GPT-4o can also be improved by OmniSearch.
>
> | Type           | fast  | slow  | never | ≤ 2-hop | \> 2-hop | no    | yes   | zh    | en    | all   |
> | -------------- | ----- | ----- | ----- | ------- | -------- | ----- | ----- | ----- | ----- | ----- |
> | GPT-4o         | 21.71 | 33.22 | 42.4  | 33.78   | 33.83    | 35.25 | 32.81 | 32.14 | 35.5  | 33.79 |
> | GPT-4V         | 17.63 | 27.8  | 40.82 | 30.8    | 28.74    | 31.71 | 29.26 | 26.44 | 34.18 | 30.25 |
> | OmniSearch (G) | 44.04 | 49.58 | 54.45 | 50.38   | 49.06    | 50.49 | 49.73 | 46.96 | 53.21 | 50.03 |
>
> You might also be referring to reasoning-based OpenAI o1, which was released concurrently with our submission. While it does not support multimodal input and cannot handle multimodal QA, highlighting the urgent need for advancements in multimodal RAG agent. Additionally, some commercial search engines in our experiments, such as Gemini, do incorporate reasoning processes.
>
> --------
>
> - *how the AI researchers are selected and trained, and the defination of the classification distribution*
>
> Answer: The definitions of the classification labels can be found in lines 170-173 of Section 3.1. We chose graduate students with at least 2 years of research experience as annotators and provided them with detailed annotation instructions.
>
> ---------
>
> - *Figure 3 typo: "on" should be "of"*
>
> Answer: Sorry for the typo. We will correct it and double-check the entire paper.

---

> ### Author Response · Authors · 2024-11-24
> **Inquiry Regarding Review Feedback for Paper Revisions**
>
> Dear Reviewer,
>
> Understanding your busy schedule, we wanted to check if the revisions and details provided have addressed your concerns. Please let us know if any issues remain or further clarification is needed. Your feedback is invaluable to us for improving our work.
>
> Thank you for your attention.

---

> ### Author Response · Authors · 2024-11-27
> **Friendly reminder to reviewer 6Ev1**
>
> Thank you very much for your valuable suggestions and comments. Based on your advice, we have **added a latency analysis of our method and a performance comparison with GPT-4o**. We have also **further discussed the dataset size**. We hope these analyses address your concerns.
>
> Considering that the rebuttal discussion phase of ICLR is about to end, we sincerely hope to receive your further feedback so that we have the opportunity to continue our discussion with you. Once again, thank you for your hard work and selfless help.

---

> > ### Comment · Reviewer_6Ev1 · 2024-12-02
> > **Thanks for the authors' addressment**
> >
> > I think the authors have addressed my concerns and questions by doing additional studies of the effectiveness of OmniSearch, I will raise my score and hope the authors can enhance the Dyn-VQA dataset in scale and more language/cultural support in the future.

---

> > > ### Author Response · Authors · 2024-12-02
> > >
> > > Thank you for your feedback and for raising the score. This has been very motivating for us. We will definitely consider enhancing the Dyn-VQA dataset with more scale and language in the future.

---

### Author Response · Authors · 2024-11-21
**General Response to Reviewers**

Thank you for your hard work and valuable comments, which make our paper better. According to ICLR's rebuttal policy, **we have revised our paper according to your comments and uploaded it to OpenReview**. In addition, for your convenience, we have **highlighted the revisions in blue in the revised paper**. Thank you again for your work and look forward to further communication with you.


Wishing everyone all the best!


Authors of Paper 8886

---

### Author Response · Authors · 2024-12-04
**Summary of Rebuttal Period and Thank You**

Dear Reviewers and Area Chairs,

Thank you all for your valuable feedback and thoughtful responses during the rebuttal period. Your insights have been instrumental in improving our paper. We have worked hard to address your concerns through additional experiments and theoretical analysis.

Meanwhile, we would like to extend special thanks to:

- Reviewer Z1ru: For **positively acknowledging our work and increasing the confidence score**. Your encouragement has been greatly appreciated.
- Reviewers 6Ev1 and ia8N: For **recognizing our responses and raising the overall rating**. Your constructive feedback has helped us refine our approach and address critical issues.

Although Reviewer t3io did not provide a response during rebuttal period, we believe that the concerns raised have been adequately addressed in our detailed rebuttal. Specifically, the primary concern shared by Reviewer t3io, as well as 6Ev1 and ia8N, regarding the dataset scale has been thoroughly discussed.

Once again, we sincerely appreciate the time and effort each of you has dedicated to reviewing our work.


Best regards,

Authors of Paper 8886

---

### Meta-Review · Area_Chair_o1Wq · 2024-12-22

**Metareview:**

The paper contributes to the field of multimodal RAGs through benchmark (Dyn-VQA) and modeling (OmniSearch). The reviewers find the problem addressed original and motivation convincing, and execution both in terms of dataset construction and experimental validation well-done. On the other hand, reviewers have concerns on the scale of the proposed dataset, and its near-future possibility of it becoming outdated due to the nature of the task. The authors argue that such a dataset is harder to construct than static datasets since it requires continual efforts to update the benchmark, which they are committed to doing. Other lower-level concerns also appear to be sufficiently addressed during the discussion period. Thus, the AC recommends acceptance.

**Additional Comments On Reviewer Discussion:**

The paper ended up with 8, 6, 6, 5 with two reviewers increased their scores to 6 after the discussion period. The reviewer with a 5 did not engage but the AC determines that their concern is sufficiently addressed.

---

### Decision · Program_Chairs · 2025-01-22

Accept (Poster)